# Opinion: Coordinated Development of Emission Inventories for Climate Forcers and Air Pollutants

Steven J. Smith[1], Erin E. McDuffie[2], Molly Charles[1]

[1] Joint Global Change Research Institute, Pacific Northwest National Laboratory, College Park, MD, USA.

[2] AAAS Science and Technology Policy Fellowships, Washington, DC, USA.

*Correspondence to*: Steven J Smith (ssmith@pnnl.gov), Erin McDuffie (erin.mcduffie@wustl.edu)

**Abstract.** Emissions into the atmosphere of fine particulates, their precursors, and precursors to tropospheric ozone, not only impact human health and ecosystems, but also impact the climate by altering Earth's radiative balance. Accurately quantifying these impacts across local to global scales historically, and in future scenarios, requires emission inventories that are accurate, transparent, complete, comparable, and consistent. In an effort to better quantify the emissions and impacts of these pollutants, also called short-lived climate forcers (SLCFs), the Intergovernmental Panel on Climate Change (IPCC) is developing a new SLCF emissions methodology report. This report would supplement existing IPCC reporting guidance on greenhouse gas (GHG) emissions inventories, currently used by inventory compilers to fulfill national reporting requirements under the United Nations Framework Convention on Climate Change (UNFCCC) and new requirements of the Enhanced Transparency Framework (ETF) under the Paris Agreement starting in 2024. We review the relevant issues, including how air pollutant and GHG inventory activities have historically been structured, as well as potential benefits, challenges, and recommendations for coordinating GHG and air pollutant inventory efforts. We argue that, while there are potential benefits to increasing coordination between air pollutant and GHG inventory development efforts, we also caution that there are differences in appropriate methodologies and applications that must jointly be considered.

## 1 Introduction

Anthropogenic emissions of greenhouse gases and air pollutants into the atmosphere have wide-reaching impacts that span local to global scales. For example, emissions of short-lived air pollutant species such as sulfur dioxide ($SO_2$), carbonaceous aerosols, and other products of incomplete combustion or fugitive sources cause enhanced levels of fine particulate matter and surface level ozone ($O_3$), both of which are harmful to human health and alter ecosystems (Mannucci et al., 2015; Malley et al., 2017; US EPA, 2020). In contrast, emissions of greenhouse gases (GHG), such as methane ($CH_4$) and carbon dioxide ($CO_2$) are relatively longer-lived in the atmosphere and alter the Earth's radiative balance, termed radiative forcing, leading to anthropogenic climate change (Myhre et al., 2013).

These classifications, however, are not always distinct as the same compounds that contribute to poor air quality can also impact Earth's radiative budget. For example, ambient fine particulate matter is the world's leading environmental health risk factor, responsible for roughly 4 million deaths worldwide in 2019(GBD 2019 Risk Factor Collaborators, 2020), but also impacts Earth's radiative balance, historically with a net cooling effect (Naik et al., 2021). This cooling effect is sufficiently strong that historical climate patterns cannot be reproduced without including these effects in models (Barnett et al., 1999; Stone et al., 2009). In addition, $O_3$ in the troposphere, enhanced in the presence of sunlight and precursor emissions, is also damaging to both environmental and human health, and is a GHG that contributes to anthropogenic climate change. Emissions of these compounds and their precursors rapidly increased during the 20[th] century (Hoesly et al., 2018), but in recent decades have both increased and decreased, depending on the specific region and individual compound (O'Rourke et al., 2021).

Inventories of air pollutant and GHG emissions help to better understand, quantify, and manage the net impacts of these compounds, thereby serving as a central tool for scientific and policy activities. Inventories quantify the rate of emissions of individual compounds into the atmosphere from specific processes or activities. They are generally based on bottom-up calculations that require detailed information on emission factors and activity data, such as fuel consumption or vehicle miles traveled, but can also incorporate direct measurements or results from physical process models. Scientific computational models rely on these inventories as inputs in order to simulate and study atmospheric processes and global change. Emission inventories that are complete (include emissions from all major sources and regions), accurate, source-specific, and developed using consistent, transparent, and comparable methodologies also provide fundamental data to inform policy decisions and allow for the evaluation of mitigation activities. For instance, air pollutant and GHG emission inventories can be used within a suite of tools to identify activities that lead to unhealthy air pollution levels or to inform the design of policies intended to mitigate air pollution or climate change.

While having similar applications, the development of national air pollutant and GHG inventories have been historically distinct. For air pollutant emissions, inventories have historically been developed by individual governments to support national and multilateral efforts to reduce air pollutant levels. While there is currently no globally consistent framework for reporting national inventories of air pollutant emissions, national and regional inventory activities have been underway since the 1970s in North America (e.g., US Environmental Protection Agency) and Europe (e.g., EMEP/European Environment Agency), with air pollutant inventory activities now well established in many countries around the world. As a result, in large part due to air pollution control efforts informed by these inventories, emission levels for most air pollutants have been generally decreasing in North America and Europe since the early 1980s, which has been broadly reflected in both inventory and observational data (Hoesly et al., 2018; Xing et al., 2015; Hand et al., 2012). Air pollution control actions in China are now also resulting in substantial emissions reductions for some air pollutants (Zheng et al., 2018b), although a comprehensive official air pollutant emission inventory has not yet been developed.

In contrast to air pollutant inventories, national efforts to quantify emissions of GHGs have been developed more recently and largely follow more standardized methodologies. Under the United Nations Framework Convention on Climate Change (UNFCCC; https://unfccc.int) 197 signatories

agreed to report annual inventories of their national anthropogenic GHG emissions and sinks using comparable methodologies. This agreement was a fundamental step in facilitating multilateral
cooperation to address climate change. Under the UNFCCC, annual emission inventory reporting is mandatory for "Annex I" countries[1] starting with emissions in 1990 up to the current year minus two, and requires that countries use IPCC Guidelines for National Greenhouse Gas Inventories (IPCC, 2006) to develop emission estimates. The first national inventories were submitted to the UNFCCC in the early 1990s. More recently as part of the reporting requirements of the Enhanced Transparency
Framework (ETF) under the Paris Agreement, Annex I countries are still required to report annual anthropogenic GHG inventories, with all other signatories having new requirements to submit national GHG inventories at least every two years, starting in the year 2024. All countries will be required to report emissions following 'good practice' GHG methodologies detailed in the 2006 IPCC GHG methodology reports.

As a follow-on to the IPCC GHG methodology guidelines and to address uncertainties in the emissions and impacts of air pollutant short-lived climate forcers (SLCFs) , the IPCC recently directed its Task Force on National Greenhouse Gas Inventories (TFI) to produce an IPCC methodology report on SLCFs (Decision IPCC-XLIX-7, May 2019). The addition of this report is intended to facilitate the estimation of both national air pollutant and GHG emissions. The TFI is currently gathering information
on existing air pollutant inventory methods, in consultation with international inventory experts, to help inform an outline for the methodology report. Pending approval of the outline by the Panel, the development of the full SLCF methodology report is currently planned to commence during the seventh IPCC assessment report (AR7) cycle.

In this perspective, we provide a background and discussion on the similarities and differences between
inventories of air pollutant and GHG emissions (Figure 1). For both air pollutants (Section 2) and GHGs (Section 3), we first highlight the importance of these compounds by discussing their health and environmental impacts. We then summarize differences in the historical use and development of GHG and air pollution emission inventories and discuss their distinct data needs and emission uncertainty considerations. In the fourth section, we discuss unique characteristics of "air pollutant SLCF
inventories" as well as approaches and considerations for quantifying the net health and climate impacts of these compounds. In the last section we argue for the benefits of developing a coordinated framework to jointly estimate GHG and air pollutant emission inventories, but also discuss the associated challenges and cautions. We end with a summary of recommendations for organizations, compilers, and research groups interested in such activities.

---

[1]  Annex I Parties include the industrialized countries that were members of the OECD (Organization for Economic Co-operation and Development) in 1992, plus countries with economies in transition (the EIT Parties), including the Russian Federation, the Baltic States, and several Central and Eastern European States.

## 2 Air Pollutant Emissions

### 2.1 Impacts

Understanding the net impacts, and mitigation opportunities, of air pollutants and GHGs on human health and the Earth system are important drivers for emissions inventory development. For air pollutants, in addition to their climate impacts (discussed in Section 4), long-term exposure to both indoor and outdoor air pollutants (i.e., $O_3$ and particulate matter) was attributable to nearly 7 million deaths worldwide in the year 2019, corresponding to the loss of over 213 million disability-adjusted life-years (GBD 2019 Risk Factor Collaborators, 2020). This global health burden is not uniformly distributed across all countries. For example, air pollution-attributable mortalities are generally much higher in low to middle income countries (generally upper right in Figure 2). The main sources of air pollution in each region differ as well, with a much larger fraction of pollutant emissions in lower income countries originating from solid biofuel use and open burning, while dominant sources tend to initially shift to fossil fuel combustion over time as industrial activities expand (GBD 2019 Risk Factor Collaborators, 2020; McDuffie et al., 2021). As also shown in Figure 2, there has been a further shift in higher income countries away from fossil emissions as air pollution policies reduce combustion emissions, but sources such as agricultural activities or wildfires remain constant or are increasing. .

From a mitigation perspective, understanding relevant emission sources through the development of emission inventories can provide key insights. For example, a key challenge in lower income countries is to accelerate the transition to lower air pollutant levels and mortality, while also improving living standards, without unduly increasing GHG emissions. Figure 2 shows historical trajectories of air pollution attributable mortality in individual countries relative to changes in emissions of fossil and industrial equivalent particulate matter over time. These trajectories illustrate that low to middle SDI countries generally move leftward over time as economic activity expands, indicating that air pollution levels remain high as fossil fuel use increases and eventually dominates over solid biomass fuels. This means that mitigating the damages caused by air pollutant and precursor emissions is likely to have a high priority, particularly for lower income countries (upper right) where air pollution impacts are high, but fossil-emissions, including GHG emissions, are still relatively low. As incomes continue to increase and once control policies are instituted, countries generally move downward in Figure 2 to lower levels of air pollution and attributable deaths (1-8% in high SDI countries after 2015). Inventories of relevant air pollutants may help to inform these activities that can lead to both improved living standards and lower pollutant levels. It is also important to note that any simultaneous GHG inventory activities in these cases should not detract from the development of data needed for improving air quality, which can result in substantial, near-term improvements in health and well-being.

### 2.2 Inventory Compounds

Inventories of air pollutants include emissions of compounds that are directly hazardous to human health, such as nitrogen oxides ($NO_x = NO_2 + NO$) and primary aerosol. These inventories also generally include precursors to tropospheric $O_3$ and secondary particulate matter, such as gaseous

ammonia ($NH_3$), carbon monoxide (CO), non-methane volatile organic compounds (NMVOCs), and $SO_2$.. These compounds are emitted from a wide variety of sources including fuel combustion for energy generation and transportation, agricultural practices, as well as industrial processes, solvent use, and waste disposal. As described below, emission estimation methods for each compound can vary by source and region, and in many cases require detailed source, temporal, and spatial information. Note that primary aerosol emissions in national air pollutant inventories are generally enumerated as $PM_{2.5}$ and $PM_{10}$ (particles smaller than 2.5 and 10 micrometers in diameter, respectively) to relate to health endpoints. However, for inventories intended for use in climate models primary aerosol emissions are enumerated as black carbon (BC) and organic carbon (OC), which are the components of $PM_{2.5}$ emissions that have the largest radiative impacts (e.g., Klimont et al., 2017). Some inventories include emissions of both total $PM_{2.5}$ as well as BC and OC.

## 2.3 Historical Inventory Use and Development

In contrast to GHG inventories (Section 3), there is no internationally accepted, globally consistent framework for reporting national emission inventories of air pollutants. As a result, the use and development of air pollutant emission inventories will often vary across world regions due to geographical differences in the relevant sources, available resources for emissions reporting and mitigation, and the relative impacts of air pollutants and their precursors.

In lower income regions, where air pollution accounted for an average of 16% of all attributable deaths in 2019 (Low and Low-middle SDI in Figure 2), the largest near-term gains in air quality will often come from large-scale transformations focused on reducing open burning (agricultural field burning, forest and grassland burning) and the household use of traditional solid biofuels and coal. In these countries, however, the availability of detailed inventory data is not necessarily a pre-requisite for effective large-scale action. In fact, even in Europe and the U.S., early air quality policies preceded coordinated inventory efforts, with official national inventory systems developed later to support and strengthen policy activities.

China provides a more recent and interesting example in this respect. Over the last two decades, there was no formal government process in place for air pollutant emission inventory development. The de-facto official air pollutant inventories were instead produced by university research groups, albeit with close ties and cooperation with government ministries. Throughout this same time period, there was increased focus on improving air quality in China, with strict controls on electric power plants and, more recently, large industrial boilers, and emission standards for new vehicles (Zheng et al., 2018a; China State Council, 2013). The result was decreasing $SO_2$ and $NO_x$ emissions by the mid-2010s (Zheng et al., 2018a). In this example, a national air pollutant inventory was not a prerequisite for quite substantial mitigation and instead, the publication of an air pollutant emissions inventory in 2018 illustrated, in retrospect, a significant declines in emissions and helped to evaluate the impact of these controls after many of these ambitious policy actions had already taken effect (Zheng et al., 2018a).

In high income countries (high SDI), where air pollution accounted for an average of 3% of all attributable deaths in the year 2019 (left in Fig 2), the use of air pollutant inventories is more central to

the policy process. Improving air quality is often a more nuanced process in these countries where significant air pollution controls are already in place. Here, emission inventories are used as a tool to help reach compliance goals, guide mitigation actions, and as input data for air quality modeling. The ultimate regulatory end points are measured pollutant concentrations, which are required to be below specified thresholds for a designated period for a region to be considered in regulatory compliance. A

key issue is also the transportation of local emissions and secondary pollutants downwind and the potential for localities within a country to have differing compliance status. As a result, detailed air pollutant inventories are regularly produced, often with a focus on higher levels of spatial and temporal detail, including characterization of emissions during short periods where air pollution exceeds regulatory limits. Indeed, the term "inventory" in the air pollution community is often taken to be

synonymous with these detailed spatial emissions data. Emission inventories, however, are only one part of regulatory, mitigation, and modeling activities, which must also contend with uncertainties in the split between natural and anthropogenic emission sources, as well as our evolving understanding of atmospheric processes that lead to the production of pollutants, their deposition, and downwind transport.

In high SDI countries, air pollutant inventories also can be used in air pollutant trading or emissions cap programs across jurisdictional boundaries, particularly for the acidifying sulfur and nitrogen dioxide species. In large part because of these regulatory programs, emissions for large sources are directly measured in the United States and Canada, and it is these measurements that form the basis of subsequent inventories for these sources instead of bottom-up calculations. In Europe, country-level

emissions targets have been negotiated under the Gothenburg protocol, making inventory estimates central to this process as well (Kuklinska et al., 2015; Sliggers and Kakebeeke, 2004; Hashimoto, 1989).

In contrast to other regions, there has been a substantial international effort across Europe to standardize methods for air pollutant emission estimation and reporting in support of the Convention on Long-

Range Transboundary Air Pollution (LRTAP) and the 2012 amendment of the Gothenburg Protocol. In 1979 the LRTAP was the first multilateral agreement to address transboundary air pollution, which created a regional framework for reducing transboundary pollution across Europe and North America. To support and standardize air pollutant emissions reporting, the *EMEP/European Environmental Agency (EEA) Guidebook* (European Environmental Agency, 2019), formerly known as the

*EMEP/CORINAIR Atmospheric Emission Inventory Guidebook*, was published in 1996 for intended use for Parties by the LRTAP. The EMEP/EEA Guidebook has been regularly updated since, most recently in 2019, to account for evolving information on emission factors and the sources of air pollutant emissions and relevant compounds (e.g., adding emission estimation guidance on black carbon in 2013 in response to the inclusion of black carbon in the 2012 Gothenburg Protocol). There has also been an

effort to align the EMEP/EEA Guidebook with the IPCC GHG nomenclature and approaches (e.g., tiered approaches), so that air pollutant emissions reporting at the country level in Europe could align with IPCC sectoral categories. As part of this alignment, however, some categories included in previous EMEP reporting protocols, which would be useful for more detailed estimation of air pollutants, have been dropped from current reporting requirements because these were not part of the IPCC protocol. In

addition, differences in emission source definitions and methodologies between the EMEP/EEA Guidebook and IPCC GHG Guidelines persist, highlighting some of the challenges with coordinating or harmonizing efforts across GHG and air pollutant emission inventories (see Section 5).

## 2.4 Data Needs

The development of emissions data used as part of regulatory processes in high SDI countries is often
supported by ministries and consultancies that specialize in the production of air pollutant emission inventories. Further, these inventories are made possible by the availability of voluminous ancillary data such as statistics on energy consumption, road networks, vehicle counts along highways, and economic and industrial activity (e.g., US EPA, 2021a; European Environmental Agency, 2019). Many country inventories rely heavily on extensive reporting processes by states/provinces and individual facilities.
Because emission factors for many air pollutants depend on operational conditions, including the performance of emission control devices, there are substantial data and analysis needs for accurate air pollutant emission estimates.

Inventory data needs and development processes in high SDI countries contrast with lower income countries where the technical infrastructure and resources needed for inventory development are often
limited, and the data needed to support detailed inventory development may not exist. For example, to estimate emissions from road vehicles, either the amount of fuel consumed, or vehicle-distance traveled by different vehicle categories (freight trucks, passenger cars, buses, etc.) needs to be known. This information is not available for many countries.

In light of these data and resource limitations, the type of information most needed to inform air quality
policies should be prioritized based on the specific policy context, resources, and circumstances in each country or entity. In low-income countries, for example, ongoing efforts to improve energy access by providing on-demand lighting (broadly comparable to the goals of the rural electrification program in the late 1930's in the United States) and alternatives to traditional cooking fuels will reduce air pollution exposure and lower the time burden of solid biofuel collection, thereby improving livelihoods
and lowering mortality rates. In this case, a key metric for progress might be prioritizing the collection of data related to the use of fuels contributing to poor air quality, (e.g., traditional solid fuels, kerosene for lighting), their trends, and their cleaner replacements (e.g., electricity for lighting, liquified petroleum gas for cooking, and use of improved cookstoves).

Further, given limited resources, inventory development will need to be prioritized based on dominant
local sources and current trends. While similar in concept to the IPCC key category analysis for GHG emissions (see below), prioritization can be more complex for air pollutants given their impacts on climate and air quality, complex feedbacks, and dependence on local conditions and rapidly changing sectors and technologies. For example, China and India are rapidly adopting road vehicle emission standards similar to those in North America and Europe, which could result in a faster movement to
higher vehicle emission control levels in these countries. This stands in contrast to the process in the United States and Europe that occurred over decades, where increasingly strict standards proceeded together with technological developments to facilitate improved emission control devices. While

transportation continues to be a dominant source of GHG emissions, if most vehicles become compliant
with higher control standards and air pollutant emissions from road vehicles are sufficiently reduced
over the coming years, this sector could become less relevant for the development of new air pollution
mitigation policies in these countries and air pollutant inventory activities could be reprioritized to focus
on other major emission sources. Though transportation inventories will continue to be important to
track compliance and usage rates[2], these rapid technological changes suggest that resources for future
air pollutant inventory improvements may best be focused on sectors and subsectors (*e.g.*, off-road
mobile sources in the USA) or specific emitted compounds (*e.g.,* NMVOCs in China) that have recently
become relatively larger sources of air pollution.

## 2.5 Uncertainty

There has been limited focus on estimating uncertainties in air pollutant inventories used by regulatory
agencies. Particularly as emission levels fall, greater focus on estimating uncertainty will be helpful to
better understand where estimated emissions may need improvement, given that incorrect emissions
data could mischaracterize trends or lead to sub-optimal policy choices. As an example, observational
and modeling analyses have indicated that there appear to be biases in United States and European air
pollutant inventory estimates for mobile sources that need to be addressed (Carslaw et al., 2011;
Anenberg et al., 2017; Anderson et al., 2014; Hassler et al., 2016), even with the extensive emission
estimation processes currently in place. This is indicative of the difficulty of making accurate estimates
for some emission sectors and highlights the need to review the detailed assumptions and procedures
used to estimate these emissions, understand sources of uncertainty, and to make use of observational
data for inventory evaluation.

Qualitatively, emission uncertainty is often higher in lower-income countries due to gaps in activity data
and a lack of emission factor information appropriate for local technologies and usage patterns.
Quantifying emissions from household solid biofuel use, for example, has proven to be particularly
difficult with large uncertainties. Not only is residential biofuel consumption uncertain, with current
global energy datasets populated with model estimates for many lower-income countries, but laboratory
emission factor measurements do not replicate varied real-world operational conditions, and in-situ
measurements are difficult to undertake. While this results in a significant level of uncertainty in
pollutant emissions from this source, given the solid evidence of the resulting impacts on human health,
this uncertainty does not detract from the overall rationale for expanding energy access. Therefore,
while emission uncertainty needs to be recognized, uncertainty is not necessarily an impediment to
effective pollution mitigation action.

---

[2] Tracking usage rates for older vehicles will be increasingly important since once the majority of
vehicles are equipped with strong emission controls, road vehicle emissions will be increasingly
impacted by super-emitters (Carslaw et al. 2011).

## 3 Greenhouse Gas Emissions

### 3.1 Impacts

The context changes significantly when considering greenhouse gases rather than air pollutant compounds. Most GHG emissions become well-mixed into the atmosphere which means that the location of emissions is largely decoupled from their impact on the Earth system. The time horizon for climate change is also relatively longer than for air pollution, given both the global scope of fossil-fuel consumption and the century to millennial atmospheric time scales of the carbon-cycle. This means that the impacts of GHG emissions and concentrations are a global and long-term issue. Further, in contrast with air pollutants, while emission estimates derived from atmospheric concentrations of GHGs can help inform inventory efforts[3], concentrations of GHG are not particularly useful as near-term regulatory end points, as concentrations change slowly and are influenced by short-term variations in the carbon-cycle and atmospheric chemistry. Instead, emission inventories of GHGs are the central tools for tracking emission trends and mitigation progress, such as those submitted regularly to the UNFCCC, which are key indicators of progress toward meeting national GHG emission reduction goals.

### 3.2 Inventory Compounds

Inventories of GHGs include carbon dioxide ($CO_2$), methane ($CH_4$), nitrous oxide ($N_2O$), as well as emissions of multiple fluorinated gases (F-gases). Fossil $CO_2$ emissions are the most straightforward to quantify because emissions can be estimated from statistics on net fossil fuel consumption and fuel energy content. Fossil $CO_2$ emissions are the largest portion of anthropogenic GHG emissions both globally (72% of global GHG emissions in 2018 (Minx et al., 2021, excluding CO2 from land-use change)), and for most large countries. A robust estimate of $CO_2$ emissions, therefore, provides for a solid foundation for total GHG emission estimates.

Methane and $N_2O$ are the next most important GHGs. These emissions originate largely from either fugitive sources or through biological processes such as anerobic decomposition and soil nitrogen cycles, making them inherently more uncertain. Fugitive emissions are also challenging to estimate and have large uncertainties since conditions vary spatially and temporally and a small number of sources can dominate the total emissions within a sector. Zimmerle et al. (2015) estimate, for example, that uncertainty in fugitive $CH_4$ emissions from the transmission and storage of natural gas in the United States is between -23% and +36% (+30%/-19% if unmodeled portions of the sector are included), and that over 36% of emissions from this sector are from a small number of individual sources. Note, however, that fugitive sources are not often dominant sources of total national anthropogenic GHG emissions. As estimated by the global Emissions Database for Global Atmospheric Research (EDGAR)

---

[3] It is also possible to estimate GHG emissions from inverse modeling of concentration measurements, from either stationary platforms (e.g., towers, surface stations), mobile platforms (e.g., road vehicles, aircraft), and/or satellite retrievals. Such inversion estimates can play a role in GHG inventory development, particularly in identifying large sources and identifying regions or sectors where inventories may have large biases.

GHG inventory (Minx et al., 2021; Crippa et al., 2021) , fugitive $CH_4$ emissions from oil and gas operations were less than 20% of total national $CH_4$ emissions in 86 of the top 100 GHG emitting countries in 2015.

Fluorinated gases are the remaining category of GHGs, which have industrial uses including as refrigerants, foam blowing agents, insulators in high voltage electrical equipment, and in other processes. Because most of these gases have no natural sources (unlike other GHGs), total anthropogenic emissions of individual F-gases can be more directly estimated using atmospheric concentration measurements or inverse modeling (Montzka et al., 2018; Rigby et al., 2019; Flerlage et
al., 2021; Brunner et al., 2017), although understanding the contributing sources still largely requires source-specific emission inventories.

## 3.3 Historical Inventory Use and Development

In contrast with air pollutant inventories, the use and development of GHG emission inventories has been strongly influenced globally by international reporting requirements from the start of national
GHG inventory reporting in the early 1990s. For example, the UNFCCC (ratified by 197 countries) requires the development and reporting of accurate, complete, consistent, comparable, and transparent annual anthropogenic GHG emission and removal estimates. Emissions reporting is facilitated by detailed templates and methodological guidelines from the IPCC (i.e., IPCC inventory guidelines) (IPCC, 2006). These emission methodologies are divided into three tiers, with higher tier methods
requiring more detailed information, but also having a higher presumed accuracy. It is recommended that the higher tier methods be used for source categories that are large or quickly growing in a particular country. There is also a focus on obtaining consistent emission trends over time, given that these are a key metric for measuring progress towards GHG reduction goals. For this reason, as data or methods are revised, it is IPCC "good practice" for countries to re-estimate emissions over the entire
reporting period (base year forward). We note that this sort of past re-estimation is not always conducted for air pollutant emissions, where it is not uncommon to find discontinuities in time series due to methodological changes over time.

For parties included in Annex I to the Convention (Annex I Parties), annual GHG inventories require anthropogenic emissions and removals of GHGs ($CO_2$, $CH_4$, $N_2O$, perfluorocarbons (PFCs),
hydrofluorocarbons (HFCs), sulfur hexafluoride (SF6), and nitrogen trifluoride $(NF_3)$[4]) from five sectors (energy; industrial processes and product use; agriculture; land use, land-use change and forestry (LULUCF); and waste). The current UNFCCC reporting guidelines on annual inventories for these Parties include use of the *2006 IPCC Guidelines for National Greenhouse Gas Inventories* and subsequent refinements(IPCC 2019).

---

[4] From paragraph 33 (p.12) of UNFCCC decision 24/CP.19, *Revision of the UNFCCC reporting guidelines on annual inventories for Parties included in Annex I to the Convention*, "Annex I Parties are strongly encouraged to also report emissions and removals of additional GHGs, such as hydrofluoroethers (HFEs), perfluoropolyethers (PFPEs), and other gases for which 100-year global warming potential values are available from the IPCC but have not yet been adopted by the COP. These emissions and removals should be reported separately from national totals."

Reporting for Non-Annex I Parties (including most lower income countries) under the UNFCCC is currently less detailed and is implemented through national communications (NCs) and biennial update reports (BURs). NCs provide information on GHG inventories and are submitted every four years. BURs provide an update of the information presented in NCs, in particular on national GHG inventories and are submitted every two years as a summary of the NC or a stand-alone report. The Non-Annex I

Parties national inventories of anthropogenic emissions should include sources and removals by sinks of all greenhouse gases, to the extent national capabilities and circumstances permit, using comparable methodologies to be promoted and agreed upon by the Conference of the Parties.

In the more recent Paris Agreement, the distinction between Annex I / Non Annex I Parties is removed and in its place is the establishment of the Enhanced Transparency Framework (ETF) for all Parties

under the commitment. Under the ETF, starting in the year 2024, each Party shall regularly provide a national inventory report of anthropogenic emissions and removals of greenhouse gases, submitted using Common Reporting Tables, and prepared using good practice methodologies from the 2006 IPCC inventory guidelines, as well as information necessary to track progress made in implementing and achieving its nationally determined contribution (NDC) GHG target.

**3.4 Data Needs**

Under UNFCCC requirements, each country reports their national inventory of anthropogenic GHG emissions to the UNFCCC secretariat. Within this framework, the development of GHG emissions data in Annex I (generally high SDI) countries is often supported by ministries and consultancies that specialize in the production of GHG emission inventories. These groups, however, are not always the

same as those that lead the development of air pollutant inventories. Similar to air pollutant inventories, however, GHG inventories also rely on voluminous ancillary datasets, including energy and infrastructure statistics. As fossil $CO_2$ emissions are typically a major source of national GHG emissions, complete and accurate energy consumption statistics are key data needed for GHG emission inventories. Increasingly detailed activity and emission factor data at the sub-national and facility-level

are also required when implementing higher tier IPCC methods. It is recommended by the IPCC that a country implement higher tier methods for key categories[5] within their inventory and to use these categories to prioritize national resources for inventory improvement (including data collection, inventory compilation, and reporting). In lower-income countries, data and resource limitations present a similar challenge as those for air pollutant inventories, however the IPCC GHG inventory guidelines

provide tier 1 methods and default emission factors, with the intent of making it feasible for all countries to develop their own national GHG inventory.

Also in contrast with air pollutant inventories, the historical focus of GHG inventory reporting has been on estimating annual national level emissions. This is largely because the temporal and spatial variability of GHG emissions is not relevant to their climatic impacts. Outside of the UNFCCC

---

[5] Key categories are emission categories that have a significant influence on a country's total inventory of greenhouse gases in terms of the absolute level of emissions and removals, the trend in emissions and removals, or uncertainty in emissions and removals.

framework, GHG inventories at the state/province and city level are increasingly being produced so that jurisdictions can also track their progress toward emission reduction goals (Ibrahim et al., 2012). More spatially detailed GHG inventories can also be used to compare inventories with atmospheric observations. These inventories require increasingly detailed underlying inputs or the use of proxy datasets to allocate national or state-level inventories to finer spatial and temporal resolutions.

**3.5 Uncertainty**

The IPCC GHG guidelines also provide methodologies for estimating emission uncertainties, which are required in national inventories submitted by Annex I countries. The IPCC provides two approaches for estimating sector and compound-specific uncertainties. The choice of approach depends on the shapes of the uncertainty distributions in the underlying input activity and emission factor datasets and the
specific estimation methods used. Neither approach, however, can capture all sources of uncertainty and as with air pollutant emissions, total inventory uncertainties are likely higher in countries with limited data. Regardless, uncertainty estimates are a valuable guide to inventory development because they point to areas where data and methodological improvements are needed most. It is likely that, because GHG inventories use more homogenous methodologies as compared to air pollutant inventories,
uncertainty analysis is more tractable.

**4 Air Pollutant Short-Lived Climate Forcers**

A major link between air pollution and climate change is through the emissions of Short-Lived Climate Forcers (SLCFs), which are generally defined as aerosols, tropospheric $O_3$ precursor compounds, $CH_4$, and shorter-lifetime hydrofluorocarbons (HFCs). There is, therefore, no "SLCF inventory" that is
distinct from air pollutants or GHGs, given that all conventional air pollutant emissions are also SLCFs. As a result, "SLCF inventory" development and data needs are already reflected in the air pollutant emissions section above.

In this section, we instead focus on those SLCFs that impact both air quality and climate, which we call 'air pollutant SLCFs'. While the term "air pollutant" can include additional categories of emissions such
as heavy metals and toxic compounds (e.g., benzene and perchloroethylene) that are "known or suspected to cause cancer or other serious health effects" (called Hazardous Air Pollutants in the U.S.),[6] we instead are discussing the specific air pollutant SLCF compounds that will be considered by the upcoming IPCC SLCF Methodology Report. These include BC and OC, as well as $SO_2$, $NH_3$, $NO_x$, CO, and NMVOCs, which each uniquely impact the formation of particulate matter and tropospheric $O_3$.

For these compounds, the relationship between emissions, climate, and air pollution is complex and often non-linear. For particulate matter, for example, most individual components have a net cooling effect (sulfates, nitrates, and OC), while BC (or soot) has an opposite warming effect. Therefore, while atmospheric aerosols are estimated to have a net cooling effect on the climate, reducing emissions to

---

[6] https://www.epa.gov/haps/what-are-hazardous-air-pollutants. Accessed October 25, 2021

reduce particulate matter concentrations and improve air quality could "unmask" GHG warming that had previously been offset (Lelieveld et al., 2019). In addition, with a lifetime in the atmosphere of roughly a week, the distribution of aerosol concentrations has a strong dependance on the spatial distribution of emissions, which is in contrast with GHG concentrations.

Understanding the net impact of reactive gas emissions is even more complex. For example, when considering their net radiative impacts, emissions of $NO_x$, CO, and NMVOCs not only impact atmospheric concentrations of $CH_4$ and $O_3$, but changes in $CH_4$ concentrations will then feedback to impact background levels of $O_3$ (e.g., Fiore et al., 2008). In addition, due to non-linearities in atmospheric chemistry, changes in $NO_x$ and NMVOC emissions can either increase or decrease local $O_3$ concentrations, depending on background conditions and the relative emissions of each (McDuffie et al., 2016). In China, for example, recent reductions in $SO_2$ and $NO_x$ emissions reduced particulate matter concentrations, however, $O_3$ levels appear to have increased because emissions of NMVOCs were not reduced at the same time (Liu and Wang, 2020). In terms of the global climate, emissions of $CH_4$, CO, and NMVOCs have increased the historical forcing from both $CH_4$ and background tropospheric $O_3$ (Myhre et al., 2013; Naik et al., 2021). Increased $NO_x$ emissions, however, have decreased $CH_4$ forcing (by decreasing $CH_4$ lifetime) but have increased background $O_3$ forcing, with the decrease in $CH_4$ forcing estimated to be larger.

Tropospheric $O_3$ itself is also complex as it is both a health hazard and a GHG. While this means that reducing tropospheric $O_3$, on a global level, is a "win win" for both air pollution and climate, the impacts of $O_3$ occur across differing spatial and temporal scales. For air quality, most attention is focused on polluted areas with high seasonal and/or peak $O_3$ levels. In contrast, $O_3$ GHG forcing is not driven by areas of peak concentrations, but by the larger-scale background $O_3$ level. As mentioned above, local concentrations will largely depend on relative emissions of $NO_x$ and NMVOC precursors, while background $O_3$ is determined largely by the magnitude of $CH_4$ and $NO_x$ emissions.

An additional complication is that the forcing effects of individual air pollutant SLCFs are much more uncertain than those of GHGs. The *relative* impact of $CO_2$ concentrations as compared to $CH_4$, for example, is much better constrained (even considering recent methane forcing updates, Etminan et al., 2016) than the radiative impact of aerosols, either collectively or for its individual components. The impact of BC is particularly uncertain, in large part because of the dynamics that stem from BC heating within the atmospheric column (Stjern et al., 2017) and warming caused by darkening snow- and ice-covered areas, which is also uncertain. There are also substantial uncertainties in atmospheric chemistry calculations of the net impact of precursor compounds on $O_3$ concentrations.

To capture these complex relationships between emissions, climate, and air pollution, and to inform or evaluate mitigation strategies, the climate impacts of air pollutant emissions are generally estimated by using air pollutant inventories as inputs into global atmospheric models(Stevenson et al., 2020). These are often the same models used to project the consequences of future GHG emission trajectories (O'Neill et al., 2016). Similar models are also used to examine the environmental and health impacts of these emissions, although models used for these purposes may have more detailed representations of atmospheric chemistry than those used for long-term simulations of climate change (e.g., Box 6.1 in

Naik et al., 2021; Appel et al., 2021). Compared to $CO_2$, $CH_4$, and F-gas emissions, air pollutant emissions data for both types of modeling are needed with spatial and temporal detail appropriate for these models, which have horizontal resolutions of roughly 50-100 km for global analyses (Feng et al., 2020). High resolution spatial emissions data is particularly important for analyses at regional to city-scales, although additional detail likely needs to be added to global emission datasets for finer-scale analysis (Huneeus et al., 2020).

When using these types of tools to evaluate the net impacts of a new technology or mitigation strategy, it is important to acknowledge the substantial uncertainty in the climate impacts of air pollutants. For example, many modeling studies have shown that reducing emissions from solid biofuel combustion for cooking and heating in the residential sector can improve air quality and avoid hundreds of thousands of attributable deaths in many densely populated regions throughout the world (e.g., McDuffie et al., 2021). However, the net climate impact of reducing biofuel use in buildings appears to be small, due to offsetting heating and cooling effects (e.g., BC vs OC), with a large uncertainty range that may even include a slight warming (Smith and Mizrahi, 2013). As a second example, while the climate and health effects of BC emission reductions from the transport sector are less ambiguous (at least under the assumption that diesel fuel is desulfurized) because of the predominance of BC emissions as compared to OC, there is still significant uncertainty in the overall magnitude of impacts.

In addition to uncertainties in net impacts, there are also uncertainties in the emission estimates themselves. As discussed above for air pollutant inventories, uncertainty estimates for air pollutants would be useful but are methodologically challenging in ways that can be distinct from GHG emissions. For example, for BC, OC, and NMVOC emissions, there are not only uncertainties in the magnitude of total emissions but also in the partitioning between BC and OC, and a multitude of different volatile organic compounds (termed speciation). Uncertainty in speciation adds to the uncertainty in climate and health impacts of sources that emit these compounds.

While there has recently been significant interest in incorporating air pollutants into climate mitigation efforts, emission and modeling uncertainties mean that the net climate and health impacts of air pollutant emissions may change substantially as scientific knowledge advances. These uncertainties pose challenges to integrating air pollutant and greenhouse gas mitigation efforts and need to be considered to produce robust policies.

**5 Benefits and challenges of coordinating air pollutant and GHG emission inventory activities**

As a result of wide-ranging health and environmental impacts of "air pollutant SLCF" emissions and to facilitate consistent, transparent, comparable, accurate, and complete emission estimates, the IPCC recently directed its Task Force on National Greenhouse Gas Inventories (TFI) to produce an additional IPCC Methodology Report on SLCFs (Decision IPCC-XLIX-7, May 2019). Additional SLCF-focused guidelines may help to coordinate air pollutant and GHG inventory methodological approaches and could enhance the ability to consider synergies between both air pollution and GHG mitigation strategies. In this context, we are considering inventory coordination to be the co-development of GHG

and air pollutant inventories (not necessarily by the same team) that include regularly reported anthropogenic emissions for a common set of sources and compounds, using consistent underlying activity data and methodologies, where appropriate. While there is no uniform approach to inventory development for all purposes, this coordination (provided the availability of methodological guidance) may be most straightforward in countries with more limited inventory resources, limited pre-existing inventory capacities, or for common emitting activities with similar underlying driver data for both types of emissions (e.g., power plants). However, air pollutant and GHG emissions inventory coordination may need to be considered more carefully in other cases (e.g., emissions related to land-use and land-use change) or in countries/entities with well-established inventory systems.

Additional benefits of coordinated air pollutant and GHG inventory efforts could include the development of common reporting and/or methodological standards for air pollutant emissions that do not always exist at present. For countries without well-developed emission inventory systems, it may also be more efficient to build a single coordinated system for estimating both GHG and air pollutant emissions. Recent revisions in EMEP/EEA guidance which better harmonizes air pollutant and GHG reporting could be useful in this context, although this guidance is sometimes limited with respect to technologies and practices not common in Europe. Better coordinated inventory development could also facilitate consideration of strategies to jointly reduce both air pollutant and GHG emissions from specific sectors, such as waste (Gómez-Sanabria et al., 2022). Increased coordination may also facilitate productive engagement between research groups and GHG and air pollutant inventory compilers that could lead to community-wide efforts to better align emission source definitions, methodological approaches, and facilitate additional data collection that could be used to improve emission inventories for both scientific and policy-oriented analyses (Janssens-Maenhout et al., 2015; Hoesly et al., 2018; Perugini et al., 2021). Similarly, existing workflows developed for producing spatially distributed (gridded) air pollutant inventories could facilitate the production of gridded GHG inventory data. These spatially explicit emissions are necessary for atmospheric modeling and are a useful tool for comparisons between regional and observationally informed emission estimates (e.g., Maasakkers et. al. 2016), as discussed in the 2019 Refinements to the IPCC GHG Guidelines (IPCC, 2019).

Challenges in coordinated development efforts, however, must also be considered and will largely depend on the different end uses of air pollutant (SLCF) and GHG emissions data and available resources in different countries and entities. For example, because air pollutant inventories are often used for impact and regulatory analyses, air pollutant inventory development will need to consider detailed technology-specific activity factors and emissions data, as well as the spatial and temporal variability in these underlying datasets. This is not always the case for GHG emissions, where aggregate annual national-level emission factors and national fuel consumption statistics (often updated more quickly than sub-national data) are often sufficient to track national mitigation efforts, at least for fossil $CO_2$ emissions. As air pollutant methodologies can also be more data intensive than GHGs inventories, air pollutant inventories may require additional tools, data, and resources, which can also lead to longer lag times. For example, mobile sources in the United States are estimated using the MOVES model, which requires a more detailed set of input data than is required for estimating national GHG emissions from this sector. In this example, estimating both GHG and air pollutant emissions using the same tools

could delay GHG emission estimates. More generally, as globally accepted, international reporting guidelines for air pollutants do not currently exist, new methodology development may also be required for air pollutants as existing methodologies (e.g., EMEP/EEA Guidebook) may not be relevant to the dominant emission sources, combustion conditions, or mitigation technologies used across low-SDI countries (IPCC, 2006). Therefore, different end uses and resource and time requirements present major
challenges to coordinating GHG and air pollutant inventory activities and should be considered.

Key emission categories can also differ substantially between air pollutant and GHG inventories. This presents an additional challenge as there may be different development and resource priorities between the two. For example, by 2012, off-road mobile combustion emissions in the United States (e.g., construction, domestic shipping, agriculture, etc.) were over a quarter of $NO_x$ emissions, but only 3% of
$CO_2$ emissions (O'Rourke et al. 2021). In the above example, increased effort to better quantify off-road mobile $NO_x$ emissions, including their spatial and temporal distribution, is likely justified to better understand drivers of air pollution, but would not be a high priority for a GHG inventory. Therefore, given finite resources and differing sources and trends across countries and regions, key emission categories should be evaluated separately for different air pollutant and GHG compounds, and data
needs and development efforts should be prioritized as needed.

This point also highlights the need to clearly define emission sectors, and ideally align these definitions across chemical compounds and driver data, while also accommodating different reporting conventions and legal requirements. This is not as straightforward as it sounds. Currently, air pollution reporting for "point sources" such as power plants or refineries is often at the facility level (which is also the level at
which some remote sensing validation methods can be applied), whereas IPCC GHG reporting guidelines call for separate reporting of fossil combustion, fugitive, and process emissions. As discussed in Section 2.3, sectoral differences have led to challenges in aligning international EMEP/EEA and IPCC methodologies across all sources of air pollutant and GHG emissions. One option to consider could be the expanded use of flexible reporting conventions that allow for the reporting of certain
compounds by sector, which is the level at which the most accurate information may exist, and would not require a detailed split between combustion and other processes. Where resources are sufficient to allow reporting at the most detailed level, data could always be aggregated to a less detailed level. Another option would be to facilitate the use of multiple reporting standards depending on the desired use. An example is the multiple reporting formats currently available for air pollutant emissions from
European countries from the EMEP Centre on Emission Inventories and Projections (https://www.ceip.at). An added challenge is that in addition to source category definitions, the chemical definitions (e.g., chemical speciation of aerosol components (BC, OC) and NMVOCs) across different sectors also need to be aligned. Chemical speciation is currently an active area of research (GEIA: Global Emissions InitiAtive; Huang et al., 2017) and air pollutant inventory development will
need to consider flexible guidance on appropriate emission speciation profiles as well.

Another consideration is the coordination of reported emission units. GHG emissions are commonly reported as individual compounds in units of mass (e.g., Tg) and also as "$CO_2$-equivalents", which are calculated using Global Warming Potentials (GWPs). Differing atmospheric GHG lifetimes mean that there is no unique method of equating the climate impact of emissions. For instance, there has been

exploration of numerous alternative metrics and policy frameworks focusing largely on comparing $CH_4$ and $CO_2$. Note, however, in idealized energy-economic model experiments, the exact $CH_4$ metric used does not have a large impact on long-term global policy outcomes as long as there are not too many barriers to increasing $CH_4$ mitigation over time, although there can be differing near-term and regional effects (van den Berg et al., 2015; Smith et al., 2013; Strefler et al., 2014). In contrast, for air pollutants,

it was recommended at a recent IPCC expert meeting (IPCC, 2018) that air pollutant emissions in an SLCF context continue to be reported in absolute units of mass for each individual compound and not be aggregated into $CO_2$-equivalents, as is recommended for GHGs. This is in part due to their much shorter atmospheric lifetimes (days to weeks vs decades to centuries) and complex climate and air quality feedbacks, as well as their much larger uncertainties in radiative forcing compared to well-

mixed GHGs. These differences translate to a very large range of potential comparison metrics that would, in essence, require value judgements as to what impact or effect should be represented.

A final aspect to consider, which is both an opportunity and a challenge, is emission uncertainty reporting. While challenging to produce, emission uncertainty estimates help to prioritize inventory improvements and inform the robustness of potential mitigation measures. The implementation of a

formal uncertainty analysis, however, does not guarantee all sources of uncertainty are captured or quantified. For example, as noted in the US GHG inventory and per IPCC guidelines, quantitative GHG uncertainty estimates focus only on parameter uncertainty, and do not consider structural uncertainty in models used for estimation (for example, for many agricultural emission sources) or the potential uncertainty from incorrect data reporting or missing emission sources (US EPA, 2021b)**.** In general,

correlation of errors across sectors (or countries) will also generally increase uncertainty estimates, but is difficult to estimate, so might not be considered. In a formal uncertainty analysis, it is also important to flag where default emission factors may be used because sector- or region-specific values are not available (per IPCC inventory guidance), thus implying a greater uncertainty for these sources(Solazzo et al., 2021). As discussed in Section 2, air pollutant inventories have not historically incorporated

emission uncertainty estimates and one challenge for the forthcoming SLCF Methodology Report will be specifying uncertainty guidance for the wide variety of methods used for air pollutant inventory production, which range from complex models (e.g., for mobile sources: MOVES in the USA and elsewhere, EMFAC in California, and COPERT in Europe) to direct emissions monitoring.

**6 Conclusions and Recommendations for Coordination Efforts**

Summarized in Table 1 are recommendations, particularly relevant for aligning and coordinating air pollutant and GHG emission methodologies and inventory reporting within the context of the UNFCCC and IPCC frameworks.

A key point is that, when developing an emissions inventory methodology, the end goals and uses should be kept in mind. For instance, air pollutants have substantial impacts on human health (Figure 2),

economies (Chantret et al., 2020), and ecosystems, resulting in different end uses of air pollutant emissions data compared to GHGs. Air pollutant and GHG emission inventories can also have different formal roles, for instance playing an integral part in regulatory and international policy frameworks in

some countries while in others, inventories are just one, arguably important, part of a larger toolset. It is
therefore critical, that any efforts to harmonize GHG and air pollutant emission methodologies and
reporting, particularly at the international level, do not detract from the production of emissions
information needed by local policy makers (e.g., high temporal and spatial resolution for air pollutant
emissions).

We also recognize that varying levels of coordination between air pollutant and GHG emission
activities may be appropriate given a country's national circumstances. Regional development priorities
should consider factors such as the relevant regional emission sources, emission trends, impacts, and
current inventory capabilities and needs. For example, the benefits from increased coordination may be
greater in countries that do not currently have established GHG and/or air pollutant inventory processes
or systems.

The level of coordination and amount of inventory data development will also need to be prioritized
based on local circumstances and available resources. For example, given limited resources, a regularly
updated and reported inventory focused on priority sectors may be more useful than a more
comprehensive effort that is not regularly updated or uses oversimplistic methodologies. The latter case
is an example of when caution must be taken when applying default (*i.e.,* lower tier) emission factor
assumptions for certain air pollutant emission sources. Some sources, such as fugitive emissions or
products of incomplete combustion (particularly of carbonaceous aerosols) have a strong dependence on
environmental and operating conditions. Therefore, their emission factors can vary by over an order of
magnitude. Strengthening institutional and technical capacity would be an important activity in this
example but would also require balancing limited inventory resources with local data availability and
needs. The development of the needed underlying information, such as energy consumption data,
however, is often generally valuable for public and private decision making and could be part of a larger
information infrastructure development strategy (Meso et al., 2009).

Resources for improving underlying data streams also need to be balanced with those required for
inventory compilation and reporting. In this regard, air pollutant inventory development in an SLCF
context should be flexible to be able to respond to changing priorities and context within a country or
entity. If emissions from a once key source category are reduced, then focus may need to shift to
sources that may now be relatively more important. This is reflected in current IPCC GHG inventory
guidance, which recommends the use of more detailed methods for larger or quickly growing source
categories. As we note above, the same level of methodological detail will not necessarily be needed for
both GHG and air pollutant emissions for a given sector, although greater spatial detail is generally
necessary air pollutant emissions data. For example, while IPCC guidelines for GHGs suggest
estimating emissions from fuel combustion separately from fugitive emissions, this split can be difficult
to determine for some sources and species. The highest priority is to have an accurate estimate of total
emissions from a facility rather than spend resources to obtain finer-grained detail. This may require
more flexible reporting options.

To reduce the development burden of a new national air pollutant SLCF-focused inventory, consideration could also be given to using research inventories as a starting point, such as EDGAR (Crippa et al., 2018), GAINS/ECLIPSE (Klimont et al., 2017) or CEDS (Hoesly et al., 2018; which

builds on EDGAR and GAINS). For some specific sources and emission species, such as large $CH_4$ or $SO_2$ sources, remote sensing estimates can also be useful adjuncts to bottom-up inventories (Schneising et al., 2020; Fioletov et al., 2016). If assessed to be credible for a specific country, these data sources could be used where resources are limited, and selectively modified or augmented where local circumstances indicate that the default assumptions used in these global estimates are not applicable.

Extensive reporting requirements will be challenging for countries with limited resources or those without existing institutional arrangements to support inventory development. Methods to reduce the burden of inventory development and reporting, including the use of open-source software approaches, should be explored to enable more countries to develop emissions data that would be useful for local policymaking. In this context, robust communication between inventory developers, users, and

researchers may help to prioritize limited resources for inventory improvement and to maximize the potential benefits of coordinated development and reporting of air pollutant and GHG emissions data.

To further assist coordination efforts, definitions in underlying driving data, sources, and chemical compounds should also be aligned between air pollutant and GHG inventories. It is important that this alignment does not decrease the utility of either GHG or air pollutant emissions information. In general,

increased consistency in reporting of air pollutant emissions across countries would be valuable in assessing the health or economic impacts of specific sources of air pollution, assessing the impact specific mitigation policies on both air pollutant and GHG emissions, quantifying impacts from trans-boundary air pollution, and to more generally facilitate more frequent and accurate emissions reporting across countries. As noted above, air pollutant SLCF emissions should continue to be reported in units

of mass of each compound (e.g., Tg of $SO_2$) and not be converted to units of $CO_2$-equivalents.

In terms of emission uncertainty, it would be useful to adapt the methods used for GHG uncertainty estimation to apply to air pollutant emissions where feasible. Some additional techniques may be needed, such as meta-models to characterize uncertainty in estimates from tools such as transportation sector emission models. Top-down emission estimates derived from atmospheric observations or remote

sensing can also be a useful component of inventory evaluation and uncertainty analysis for both air pollutant and GHG emissions by providing an independent estimate of total emissions, albeit with its own methodological uncertainties. This will require increased communication and coordination between inventory developer/compiler and research communities (Perugini et al., 2021).

Finally, an international expert review system for air pollutant SLCF inventories should also be

considered. Currently, intensive country reviews are incorporated into the GHG inventory reporting processes under the UNFCCC and will continue under the Paris Agreement as a means to improve inventory accuracy, quality, and transparency. Pulles (2017) has argued that, while the UNFCCC review process has resulted in inventory adjustments, there was little evidence that this process has resulted in significant changes in estimated total GHG emissions (although we note that there are likely other

benefits of the review process). Pulles (2017) and Hanle et al. (2019) have argued that the review process should be redesigned and streamlined. Such a redesign could be important as inventory

activities expand to more countries and if international inventory reporting expands to include air pollutant SLCF emissions, which would add additional burdens to the existing review system.

We presented an overview of the history and issues associated with developing both air pollution and GHG inventories, the importance and impacts of SLCFs, and the benefits and challenges of coordinating joint GHG and air pollutant emission estimates through the development of an air pollutant SLCF inventory. A more thorough assessment of the potential benefits to both policy and research communities from such coordination efforts may be useful.

## Author Contribution

SJS and EEM drafted text. MC conducted data analysis and visualization.

## Acknowledgements

SJS was supported for this work by the U.S. Environmental Protection Agency. EEM was supported by an American Association for the Advancement of Science (AAAS) Science and Technology Policy Fellowship. The authors thank Bill Irving, Vince Camobreco, Robert Pinder, and Marcus Sarofim for their comments and suggestions during manuscript development. We also thank Zbigniew Klimont, an anonymous reviewer, and Luke Western for their helpful comments during the review process.

## Code/Data Availability

All data used in this work was from publicly available sources. Data as complied for use in Figure 2 is supplied as a supplemental data file.

## Competing interests

The authors declare no competing interests

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

Table 1: Recommendations for Coordination Efforts

- Account for the different end uses of air pollutant and GHG emissions inventory data while maintaining the useability of each (e.g., consider necessary sectoral, spatial, and temporal data resolutions).

- Consider regional development priorities and recognize that different levels of coordination may be appropriate for different countries or entities.

  - Understand the key emission sources across regions and compounds to prioritize development/reporting and data collection resources.
  - Develop flexible reporting protocols and methodologies that allow for accurate reporting but that are not overly burdensome. For example, the use of the IPCC tier approach (e.g., higher tiers needed for some emission species in a particular sector, but lower tiers might be acceptable for other species).
  - Consider the use of research inventories as a starting point and enhance communication between air pollutant and GHG inventory compiler, research, and policy communities

- Align, and publicly document, definitions of sectoral, chemical speciation, and activity driver data between air pollutant and GHG inventories.

- Air pollutant SLCFs should not be reported in units of $CO_2$-equivalents.

- Draw on existing GHG inventory methodologies to provide estimates of air pollutant emission uncertainties.

- An international expert review system for air pollutant SLCF inventories should be considered.


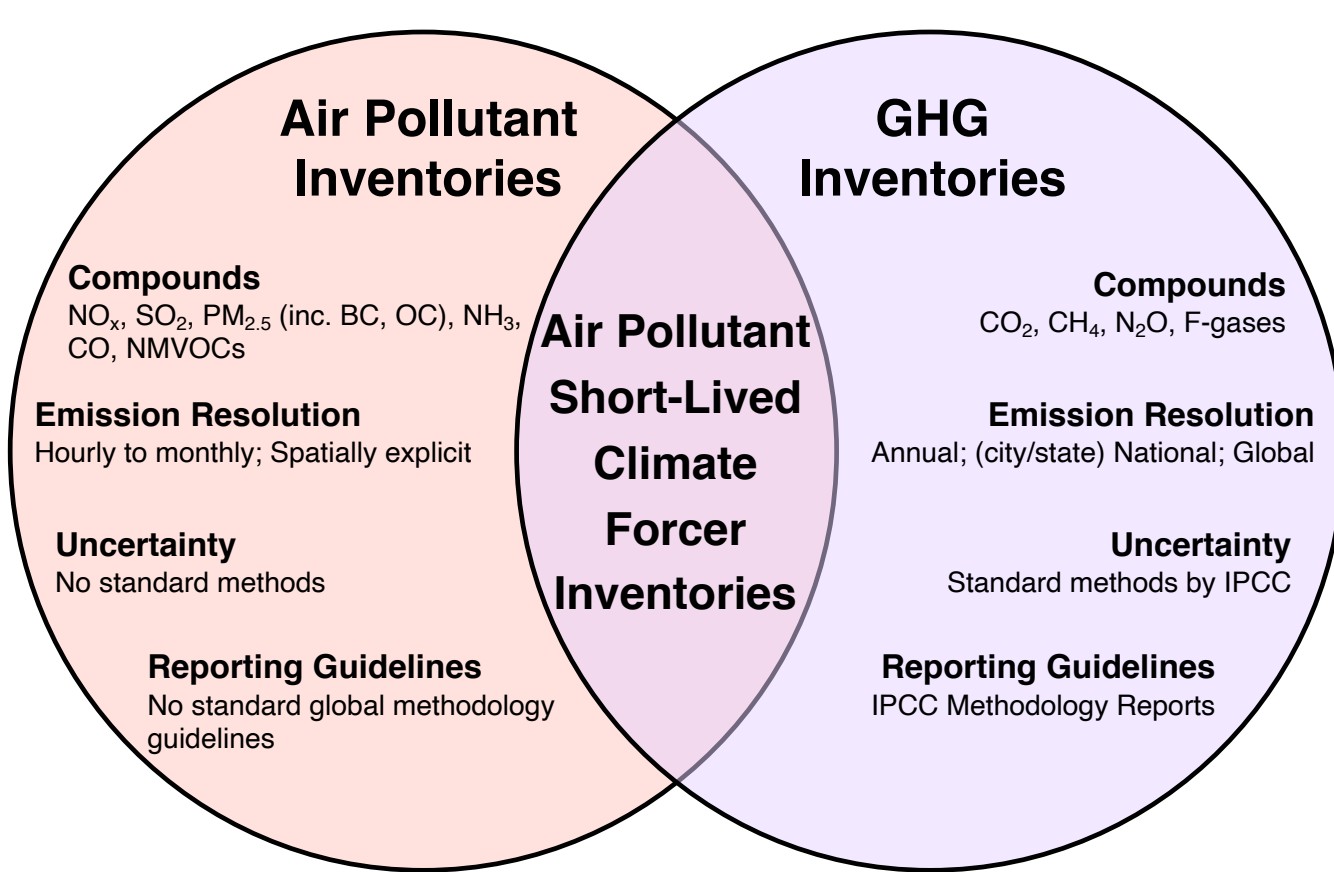

**Figure 1:** Schematic diagram illustrating the main characteristics of greenhouse gas (GHGs) vs air pollutant inventories, also highlighting overlap in characteristics inventories of Air Pollutant Short-Lived Climate Forcers (SLCF). As discussed in the main text, an "Air Pollutant SLCF inventory" is largely a matter of context rather than a distinct category of inventory data. Note also that all the air pollutants and some of the GHGs listed above are considered SLCF compounds.


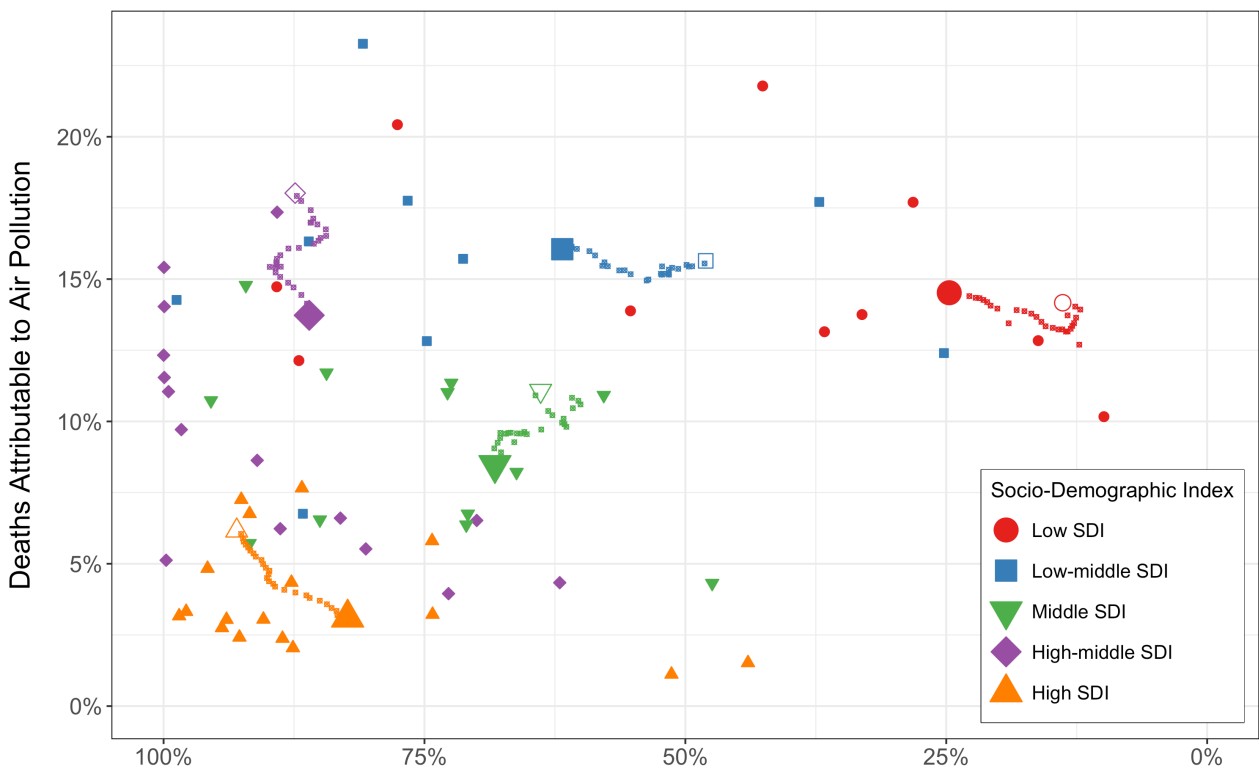

**Figure 2: Illustrative figure showing the relationship between emissions source and the fraction of 'all cause' deaths attributable to annual air pollution (e.g., outdoor/indoor PM$_{2.5}$ and O$_3$) exposure over time and by socio-demographic category. Fraction of all deaths in 2019 attributable to annual air pollution exposure (y-axis) by major Socio-Demographic Index (SDI) groups (large symbols) from the GBD project (GBD 2019 Risk Factor Collaborators, 2020) plotted against the fraction of equivalent particulate matter (PM) emissions from fossil/industrial sources (van Marle et al., 2017; O'Rourke et al., 2021; see Appendix for definitions). The largest countries within each SDI group are shown as small individual symbols. The trajectory over 1990 (open symbols) to 2019 (large filled symbols) for each country SDI group is shown by the smaller trailing symbols. Countries in the lower left position have the lowest mortality from air pollution with particulates largely from fossil and industrial sources. Countries in the upper right have a higher disease burden from air pollution and particulates that are largely from non-fossil sources such as forest and grassland fires and household use of solid biofuels.**



 **Appendix – Data sources and methodology for Figure 2**

## A.1 Equivalent Particulate Matter (PM) Emissions

Illustrating the relationship between emissions and air pollution-related mortalities, as we have done in Figure 2, is challenging because of the complexities discussed in the main text. In order to show the general sources of particulate pollution in different countries over time we developed a metric we call equivalent particulate emissions. This represents a generic estimate of how much particulate matter is produced by emissions in each country.

This metric is constructed by summing the primary BC and OC particulate emissions together with an estimate of secondary particulate formation obtained by multiplying each precursor emission species by a multiplier that represents the amount of particulate (by weight) that is produced globally by that emission species. Molecular conversions are applied as necessary. For example, each kg of $SO_2$ gas that is converted into sulfate aerosols results in 1.8 kg of ammonium bisulfuate aerosol, but we assume that only 55% of emitted $SO_2$ is converted into sulfate aerosol as the remainder is lost through wet or dry deposition (Ghan et al., 2013a). Sources for conversion factors used are provided in table A-1.

This metric is approximate given uncertainties and spatial variation in nearly all of these conversion factors, which will also change over time. For example, the role of $NH_3$ in particle formation is increasing as $SO_2$ emissions decrease. The globally representative values used here serve to provide a rough estimate of how much particulate matter is produced by emissions of each species and allow us to provide an estimate of the fraction of particulate matter originating from fossil/industrial sources as shown in Figure 2. While a more detailed calculation would change some details of the figure, we are highlighting here the large-scale features and general trends over time.

For Figure 2 we required the fraction of PM-equivalent emissions that are from fossil sources. Non-fossil/industrial emissions were considered to be the following sources: grassland burning, forest burning (including deforestation), agricultural waste burning on fields, and all combustion of solid biomass fuels. All other emissions were considered fossil sources. We have categorized all anthropogenic emissions as "fossil/industrial", including emissions from agricultural systems, which are largely, although not entirely, facilitated by fossil-fueled inputs and industrial-scale operations (e.g., fertilizer application, confined animal feeding operations, etc.). For primary OC emissions, the conversion is from carbon units, used in the inventory data, to total OC weight.

We note that secondary organic aerosols (SOA) due to anthropogenic NMVOC emissions are not included as we did not find a definitive source for this conversion (although we note that a large portion of anthropogenic SOA originates from biogenic VOC emissions).

Mineral $PM_{2.5}$ emissions, such as fly-ash from coal combustion, are also not included because these are not available in global emission inventories by country. Mineral $PM_{2.5}$ emissions are implicitly included in most air pollutant inventories as part of $PM_{2.5}$ in global and country level inventories, but generally are not explicitly estimated as a separate category. Because of this mineral $PM_{2.5}$ emissions are not included in most model studies to date (Philip et al., 2017).

The x-axis of figure 2 indicates the fraction of total PM-equivalent emissions that are from fossil and industrial sources. As noted above we consider fossil/industrial emissions to be all emissions **excluding** the following:

- Solid biomass CEDS (anthropogenic) emissions across all sectors
- 5C_Waste-incineration and open burning of waste (CEDS)
- All open burning emissions (forest fires, grassland files, ag waste burning on fields) from GFED

We recognize that there is a small portion of emissions from liquid and gaseous fuels that are derived from biofuels that would be included in the fossil/industrial faction, but these are small. Similarly,
emissions from both waste incineration and open burning result from a combination of fossil (*e.g.*, plastics) and non-fossil (paper, and food) sources.

| Emission | Emission Unit | Conversion Factor | Source |
|---|---|---|---|
| $SO_2$ | kt $SO_2$ | 0.99 | (Ghan et al., 2013b) |
| $NO_x$ | kt $NO_2$ | 0.61 | (Feng and Penner, 2007) |
| $NH_3$ | kt $NH_3$ | 0.5 | (Feng and Penner, 2007) |
| BC | kt C | 1 | NA |
| OC_biomass | kt C | 1.8 | (Klimont et al., 2017) |
| OC_fossil | kt C | 1.3 | (Klimont et al., 2017)) |

Table A-2 — Conversion factors used for the Equivalent-PM emissions calculation. The factor converts emissions in terms of weight units used in the emission inventory to weight of particulate matter
accounting for the fraction of emissions that are converted to aerosol and changes in molecular weight.

A.2 Health Impacts of Air Pollution

Health impact results used in this work were downloaded from: http://ghdx.healthdata.org/gbd-results-tool (downloaded Feb 21, 2021), as documented in GBD 2019 Risk Factor Collaborators (2020) . Shown in Figure 1 is the ratio of deaths from "Air pollution" exposure to deaths from "All_Cause". This
ratio represents the deaths that are attributable to annual exposure to total air pollution (indoor $PM_{2.5}$, outdoor $PM_{2.5}$, and $O_3$; caused by type II diabetes, stroke, chronic obstructive pulmonary disorder, lower respiratory infections, ischemic heart disease, lung cancer, and neonatal disorders [low birth weight, pre-term births]) relative to the total number of deaths caused by 369 diseases and injuries assessed by the 2019 GBD. Changes in this ratio overtime may reflect changes in both the number of deaths
attributable to air pollution (due to changes in both air pollution and baseline mortality rates), as well as trends in the remaining causes of death in particular countries.

The emissions and health impact data, as processed for Figure 2, is provided as a supplementary data file.