# Peer review of "Opinion: Coordinated Development of Emission Inventories for Climate Forcers and Air Pollutants"

_Atmospheric Chemistry and Physics, 2021_

## Author Response (AR2)

**Response to Reviewers**

We thank the three reviewers for their insightful comments. These have greatly helped us to refine and improve the clarity of our key points and to improve the quality of our discussion. We have addressed each comment in detail below. Our responses are in blue, with specific changes to the text highlighted in *blue italics*. All line numbers in this document correspond to the line numbers in the updated (tracked-changes) version of the manuscript.

**CC1: 'Comment on acp-2021-1059 regarding F-gases', Luke Western, 31 Jan 2022**

I welcome your opinion article "Coordinated Development of Emission Inventories for Climate Forcers and Air Pollutants", which I believe is timely and pertinent.

Below I raise a few issues in the discussion of F-gases, a minor aspect of the discussion, which may be misleading or confusing to the reader.

We thank Dr. Western for providing public comments regarding the discussion of HFCs in our original text. These comments have helped us improve the clarity and specificity of how we discuss emissions of fluorinated gases throughout the manuscript. Specifically, we have provided additional clarification regarding the specific fluorinated compounds that are required to be reported under the UNFCCC, in addition to other relevant minor updates. We have responded to each comment individually below.

The sentence beginning line 307, 'The Non-Annex I Parties national inventories of anthropogenic emissions should include sources and removals by sinks of all greenhouse gases not controlled by the Montreal Protocol…', may cause confusion. HFCs are controlled under the Kigali Amendment to the Montreal Protocol. The reader is therefore left with the impression that HFC emissions should not be reported by Non-Annex I Parties, which, given the discussion in the previous paragraph about Annex I reporting, appears contradictory.

Following the previous point, it is unclear to me why it is suggested that gases controlled under the Montreal Protocol should not be reported. Emissions of gases controlled under the Montreal Protocol, excluding HFCs, are currently not reported through any universal framework – only production and consumption are reported to UNEP under the Montreal Protocol.

The original language on line 309 regarding the 'Montreal Protocol' was taken directly from the UNFCCC reporting guidelines on annual inventories for Parties included in Annex I to the Convention (24/CP.19). The guidelines state, "Annex I Parties shall use the methodologies provided in the 2006 IPCC Guidelines, unless stated otherwise in the UNFCCC Annex I inventory reporting guidelines, and any supplementary methodologies agreed by the COP, and other relevant COP decisions to estimate anthropogenic emissions by sources and removals by sinks

of GHGs not controlled by the Montreal Protocol".
(https://unfccc.int/resource/docs/2013/cop19/eng/10a03.pdf#page=2)

In order to avoid confusion in this manuscript, we have removed the reference to the Montreal Protocol on line 309 since we have already defined (on line 297) the specific classes of GHGs that are required to be reported by Annex I countries under the UNFCCC.

Line 379 - The Non-Annex I Parties national inventories of anthropogenic emissions should include sources and removals by sinks of all greenhouse gases, to the extent national *capabilities and circumstances* permit, using comparable methodologies to be promoted and agreed upon by the Conference of the Parties.

It is not clear whether any of the discussion of F-gases applies to the entire suite of fluorinated gases, or only those included when following UNFCCC reporting guidelines.

We have updated the language throughout the relevant sections of text and added a new footnote to clarify that when discussing the historical development of GHG inventories, this manuscript is primarily focused on those specific F-gases that are required to be reported under the UNFCCC (per 24/CP.19). However, we have retained the original language in other places ofthe manuscript where the key points being made can apply to both currently required classes of F-gases and those that are only encouraged to be reported. For example:

Line 368 - For parties included in Annex I to the Convention (Annex I Parties), annual GHG inventories require anthropogenic emissions and removals of GHGs ($CO_2$, $CH_4$, $N_2O$, perfluorocarbons (PFCs), hydrofluorocarbons (HFCs), sulphur hexafluoride (SF6), and nitrogen trifluoride $(NF_3)$[5]) from five sectors (energy; industrial processes and product use; agriculture; land use, land-use change and forestry (LULUCF); and waste).

(New footnote) *[5]From paragraph 33 (p.12) of UNFCCC decision 24/CP.19, Revision of the UNFCCC reporting guidelines on annual inventories for Parties included in Annex I to the Convention, "Annex I Parties are strongly encouraged to also report emissions and removals of additional GHGs, such as hydrofluoroethers (HFEs), perfluoropolyethers (PFPEs), and other gases for which 100-year global warming potential values are available from the IPCC but have not yet been adopted by the COP. These emissions and removals should be reported separately from national totals."*

Line 324 - Inventories of GHGs include carbon dioxide ($CO_2$), methane ($CH_4$), nitrous oxide ($N_2O$), as well as *emissions of multiple* fluorinated gases (F-gases).

Line 346 - Because most of these gases have no natural sources (*unlike other GHGs*), total *anthropogenic* emissions *of individual F-gases* can be *more directly* estimated using atmospheric concentration measurements *or inverse modeling* (Montzka et al., 2018, Rigby et al., 2019, Flerlage et al., 2021, Brunner et al., 2017), although understanding *the contributing sources* still *largely* requires *source-specific* emission inventories.

The sentence beginning line 279 suggests that F-gases can be estimated using atmospheric concentration measurements "in some cases over large regions". It is not clear what these "cases" are and what the limiting factor is here. The limiting factor is the spatial coverage of the measurements (see e.g. Weiss et al., 2021).

We thank Dr. Western for this comment and the opportunity to clarify this statement. The intent of this sentence was not to discuss the limited availability of atmospheric measurements of F-gases (though important), but rather highlight that in comparison to other GHGs, the total anthropogenic emissions of F-gases are relatively easier to derive from direct atmospheric concentration measurements, since F-gases have no natural sources. For other GHGs, it is more difficult to derive anthropogenic contributions (UNFCCC requires reported anthropogenic emissions only) from concentration measurements since the natural background concentrations would need to be quantified and removed. In this section, we did not intend to discuss the limitations or uncertainties of HFC observations or observationally-derived HFC emissions estimates, but did add two references that provide a more thorough discussion on the topic. We have clarified this sentence as follows.

Line 346 - Because most of these gases have no natural sources (*unlike other GHGs*), total *anthropogenic* emissions *of individual F-gases* can be *more directly* estimated using atmospheric concentration measurements *or inverse modeling* (Montzka et al., 2018, Rigby et al., 2019, Flerlage et al., 2021, Brunner et al., 2017), although understanding *the contributing sources* still *largely* requires *source-specific* emission inventories.

I raise these points in what I believe is a very important opinion article only to aid future confusion and hopefully push change.

We thank Dr. Western again for the helpful and insightful comments.

Luke Western
NOAA, USA / University of Bristol, UK

Weiss, Ray F., A. R. Ravishankara, and Paul A. Newman. "Huge gaps in detection networks plague emissions monitoring." Nature (2021): 491-493.
Citation: https://doi.org/10.5194/acp-2021-1059-CC1

**RC1: 'Comment on acp-2021-1059', Zbigniew Klimont, 12 Mar 2022**

The issue of harmonization of GHG and air pollutant inventories has been discussed and attempted at regional and global level since a while...with mixed the sucess. The recognized importance of SLCFs led to another attempt of harmonizatiion with GHG inventory process; in fact IPCC tried that some 10-15 years ago but at the time the conclusion was that the uncertainties and data availability is not satisfactory; at the time the focus was on black carbon. The new attempt appears more promising and the efforst is supported by a larger group of scientists. This paper provides a good account of key advantages and challenges associated with the development of the harmonized (or coordinated) inventories that could serve several purposes, even though the process is driven by (and the needs) of IPCC.

We thank the reviewer for their very thorough and insightful comments. These comments greatly helped us to clarify the scope of this manuscript and to emphasize our key points throughout. We have responded to all comments below. In particular, we have added a section as requested that provides more background information on the LRTAP Convention and EMEP/EEA Guidebook and a discussion of previous efforts to align the EMEP/EEA Guidebook with IPCC GHG guidance and some of the associated challenges. As the reviewer also pointed out, there is no uniform approach to inventory development. Therefore, we have clarified throughout the discussion section that 'coordination' or 'harmonization' efforts that we discuss here may look very different for different countries depending on their local resources, priorities, and existing capabilities. Please see our specific responses below.

My impression is that the authors occasionally deter from the main purpose of the IPCC TF and discuss this development as a way to develop an all-purpose inventory, which in my view is neither feasible nor desirable. Such discussion appears in few places refering to the multipurpose of inventories and is inlcuded also in COnclusions (see from line 541). The key reasons and end uses for this specific work and process are clear and are not meant to replace some of the existing work and established systems like the European EEA/EMEP Guidebook and the EU or UNECE reporting system. These serve differnet purposes. I agree that learning, exchange of information, benefit from resourves available within well established national and international systems shall be utilized.

We agree with the reviewer that there is no uniform approach to inventory development for all purposes and revised the manuscript to stress that 'coordination' or 'harmonization' can look very different in different countries/regions with different capacities and priorities. We have reorganized section 6 as follows below.

In addition, while this manuscript is motivated in part by the IPCC process, we are aiming this also to a larger and more general audience with an interest in inventory development processes. For example, in many countries, there may be a higher priority on emission inventories aimed at evaluating air pollution, and we point out that this may be more important locally, and from an impacts perspective, than international reporting commitments. Therefore, it's important that any new guidelines or commitments, such as through the IPCC or UNFCCC,

not detract from what may be more urgent regional priorities. We have also clarified that, while our discussion is general and may have points relevant to any countries, we have a particular focus on countries and regions with more limited resources and limited pre-existing inventor development capability.

Line 527 - *While there is no uniform approach to inventory development for all purposes,* this coordination *(provided the availability of methodological guidance) may* be most straightforward *in countries with more limited inventory resources, limited pre-existing inventory capacities, or* for common emitting activities with similar underlying driver data for both types of emissions (e.g., power plants). *However, air pollutant and GHG emissions inventory coordination may* need to be considered more carefully in other cases *(e.g., emissions related to land-use and land-use change) or in countries/entities with well-established inventory systems.*

Line 555 - Challenges in coordinated development efforts, however, must also be considered and will l*argely depend on the different* end uses of air pollutant (SLCF) and GHG emissions data *and the available resources in different countries and entities.* For example, *because air pollutant inventories are often used for impact and regulatory analyses,* air pollutant inventory development *will* need to consider *detailed technology-specific activity factors and emissions data, as well as* the spatial and temporal variability in *these* underlying datasets. *This is not always the case for GHG emissions, where aggregate annual national-level emission factors and national fuel consumption statistics (often updated more quickly than sub-national data) are often sufficient to track national mitigation efforts, at least for fossil $CO_2$ emissions. As air pollutant methodologies can also be more data intensive than GHGs inventories, air pollutant inventories may require additional tools, data, and resources, which can also lead to longer lag times.* For example, mobile sources in the United States are estimated using the MOVES model, which requires a more detailed set of input data than is required for estimating national GHG emissions from this sector. *In this example,* estimating both GHG and air pollutant emissions using the same tools could delay GHG emission estimates. *More generally, as globally accepted, international reporting guidelines for air pollutants do not currently exist, new methodology development may also be required for air pollutants as existing methodologies (e.g., EMEP/EEA Guidebook) may not be relevant to the dominant emission sources, combustion conditions, or mitigation technologies used across low-SDI countries (IPCC, 2006). Therefore, different end uses and resource and time requirements present major challenges to coordinating GHG and air pollutant inventory activities and should be considered.*

Line 597 - *As discussed in Section 2.3, sectoral differences have led to challenges in aligning international EMEP/EEA and IPCC methodologies across all sources of air pollutant and GHG emissions.*

Line 664 - *We also recognize that varying levels of coordination between air pollutant and GHG emission activities may be appropriate given a country's national circumstances. Regional development priorities should consider factors such as the relevant regional emission sources,*

*emission trends, impacts, and current inventory capabilities and needs. For example, the benefits from increased coordination may be greater in countries that do not currently have established GHG and/or air pollutant inventory processes or systems.*

Line 1031 - Table 1 - Consider regional development priorities *and recognize that different levels of coordination may be appropriate for different countries or entities.*

The other more general comment is about lack of reference and discussion of the EEA/EMEP Emission Inventory Guidebook and the UNECE (Air Convention) supported process that lead to it. Interestingly, here the air pollutant inventory system (developped in the 90's) was extended and an attempt of harmonization with the IPCC formats was made over time, simplifying some of the air pollutatn reporting but at the cost of detaching the reporting categories from the sources of emissions, loosing important detail for NMVOCs, and making further enhancements to improve inventories of particualte matter more difficult. Neverthless, EEA/EMEP guidebook and methods continue to be developed, used by all UNECE countries and many other countries of the world, even if the methods are often not very representative for developing countries. Several developers and users of EEA/EMEP guidebook and methodology are members of the IPCC TF. I believe the paper shall include brief discussion and refernce to this, for example in section 2.3, but also in Figure 1, where one coudl mention the only internationaly (not globally) accepted reporting system and methods (beyond scientific tools and attepts, like EDGAR or CEDS and some specific models).

We thank the reviewer for the additional context and for highlighting this very important omission. We have added additional background and discussion of the EMEP/EEA Guidebook in Section 2.3, as suggested, and clarified our original statement at the beginning of this section.

Line 165 - In contrast to GHG inventories (Section 3), there is no *internationally accepted,* globally consistent framework for reporting national emission inventories of air pollutants.  As a result, the use and development of air pollutant emission inventories will often vary *across world* regions due to geographical differences in the relevant sources, available resources for emissions reporting and mitigation, and the relative impacts of air pollutants and their precursors.

Line 214 - *In contrast to other regions, there has been a substantial international effort across Europe to standardize methods for air pollution emission estimation and reporting, in support of the Convention on Long-Range Transboundary Air Pollution (LRTAP) and the 2012 amendment of the Gothenburg Protocol. In 1979 the LRTAP was the first multilateral agreement to address transboundary air pollution, which created a regional framework for reducing transboundary pollution across Europe and North America. To support and standardize air pollutant emissions reporting, the EMEP/European Environmental Agency (EEA) Guidebook (European Environmental Agency, 2019), formerly known as the EMEP/CORINAIR Atmospheric Emission Inventory Guidebook, was published in 1996 for intended use for Parties by the LRTAP. The EMEP/EEA Guidebook has been consistently updated since, most recently in 2019, to account for evolving information on the sources of air pollutant emissions and relevant compounds (e.g.,*

*adding emission estimation guidance on black carbon in 2013 in response to the inclusion of black carbon in the 2012 Gothenburg Protocol). There has also been an effort to align the EMEP/EEA Guidebook with the IPCC GHG nomenclature and approaches (e.g., tiered approaches), so that air pollutant emissions reporting at the country level in Europe could align with IPCC sectoral categories. As part of this alignment, however, some categories included in previous EMEP reporting protocols, which would be useful for more detailed estimation of air pollutants, have been dropped from current reporting requirements because these were not part of the IPCC protocol. In addition, differences in emission source definitions and methodologies between the EMEP/EEA Guidebook and IPCC GHG Guidelines persist, highlighting some of the challenges with coordinating or harmonizing efforts across GHG and air pollutant emission inventories (see Section 5).*

*Line 539 - Recent revisions in EMEP/EEA guidance which better harmonizes air pollutant and GHG reporting could be useful in this context, although this guidance is sometimes limited with respect to technologies and practices not common in Europe.*

I guess, the authors have spent some time disussing the order in the paper and decided for air pollutants, GHG, SLCF and i was surprised but the start of the discussion with the air pollution impacts on health as entry point to this paper - is this because of the attempt to make the link to SDGs and Paris Agreement? I was wondering myself what woudl be best. Considerig the fact that this the IPCC driven process that attempts to bring and update methods to include SLCFs along the GHGs, I woudl probably start with GHG section followed by the SLCF discsussion that woudl link to air pollutant inventories and experience as this includes pretty much all SLCFs, apart from CH4 and HFCs that are already included under GHG inventories. I was also wondering if the 'impact' sections in air pollutant and also GHG sections shall not be part of the introduction to the paper giving background to inventoies in general.

We thank the reviewer for their suggestion. Indeed, there is no unique way to organize this material. We have decided to maintain the existing structure to the manuscript (e.g., air pollutants (Section 2), GHGs (Section 3), SLCF (Section 4)). We have decided to start with a distinct section on air pollutants emissions so that we can discuss the longer use and development history of air pollutant inventories that has been, until very recently, distinct from the more recent IPCC focus on air pollutants that are also SLCFs. We follow this section with a separate section of GHG emissions so that we can more easily compare and contrast these historical use and development efforts with air pollutant inventories. We use the SLCF terminology in Section 4 to align with the terminology of the IPCC and to focus on the specific compounds considered in the ongoing IPCC effort.

However, we have clarified the structure within the each section, specifically the air pollutants impacts section where we have discussed the important connection between impacts and inventories.

Line 98 - For both air pollutants *(Section 2)* and GHGs *(Section 3), we first highlight the importance of these compounds by discussing* their health and environmental impacts. *We*

*then summarize* differences in the historical use and development of GHG and air pollution emission inventories and discuss their distinct data needs and emission uncertainty considerations. In the *fourth* section, we discuss unique characteristics of "*air pollutant* SLCF inventories" as well as approaches and considerations for quantifying the net health and climate impacts of these compounds.

Line 110. *Understanding the net impacts (and mitigation opportunities) of air pollutants and GHGs on human health and the Earth system are important drivers for emissions inventory development. For air pollutants, in addition to the climate impacts (discussed in Section 4),* long-term exposure to both indoor and outdoor air pollutants (i.e., $O_3$ and particulate matter) was attributable to nearly 7 million deaths worldwide in the year 2019, corresponding to the loss of over 213 million disability-adjusted life-years (GBD 2019 Risk Factor Collaborators, 2020).

Finally, I am not entirely sure what purpose Figure 2 serves - in the context of this paper. Maybe additional reason for this question is my personal believe that the concept of euivalent particulate matter emissions is not very useful and not only bacuse of the spatial scales at which air pollution is truly relevant, rather than large scale (national or regional averages) but also because of changing atmosphere composition the assumed relationships on the roles of each species contributing to ambient PM are far from constant.

The link between the figure and the paper has been made clearer in the revised version with the following text.

We note that equivalent PM emissions metric is just used to derive the fraction of fossil/industrial PM for the x-axis for purposes of illustration. We recognize that this is not precise (and have highlighted this), but it is not necessary for this to be a precise calculation to illustrate the broad trends between regions and over time. Broadly speaking, incorporating different regional emission-concentration relationships would shift relative locations slightly, but would not change the overall message that is being conveyed or the points we have raised based on this figure.

Line 127 - *From a mitigation perspective, understanding relevant emission sources through the development of emission inventories can provide key insights. For example, a key challenge in lower income countries is to accelerate the transition to lower air pollutant levels and mortality, while also improving living standards, without unduly increasing GHG emissions. Figure 2 shows* historical trajectories *of air pollution attributable mortality in individual countries relative to changes in emissions of fossil and industrial equivalent particulate matter over time. These trajectories* illustrate that low to middle SDI countries generally move leftward over time as economic activity expands, *indicating that* air pollution levels remain high as fossil fuel use increases and eventually dominates over solid biomass fuels. *This means that mitigating the damages caused by air pollutant and precursor emissions is likely to have a high priority, particularly for lower income countries (upper right) where air pollution impacts are high, but fossil-emissions, including GHG emissions, are relatively low.* As incomes continue to increase and once control policies are instituted, countries generally move downward in Figure 2 to

*lower levels of air pollution and attributable deaths (1-8% in high SDI countries after 2015). Inventories of relevant air pollutants may help to inform these activities that can lead to both improved living standards and lower pollutant levels. It is also important to note that any simultaneous GHG inventory activities in these cases should not detract from the development of data needed for improving air quality.*

Few more specific comments:

line 24: one coudl say 'grennhouse gases and air pollutants' instead of 'gases and particultes'

Changed

Line 24 -  Anthropogenic emissions of *greenhouse gases and air pollutants* into the atmosphere have wide-reaching impacts that span local to global scales.

line 26: for completness, fugitive sources shall be mentioned too, not only incomplete combution, since NMVOC, NH3, and even fugitive PM play a role in formation of pollution

Changed

Line 25 - For example, emissions of short-lived air pollutant species such as sulfur dioxide ($SO_2$), carbonaceous aerosol, and other products of incomplete combustion or *fugitive sources* cause enhanced levels of fine particulate matter and surface level ozone ($O_3$), both of which are harmful to human health and ecosystems (Mannucci et al., 2015, *Malley et al., 2017, EPA, 2020).*

line 41: I'd say 'recent decades' rather than 'recent years'

Changed

Line 41 - Emissions of these compounds and their precursors rapidly increased during the 20[th] century (Hoesly et al., 2018), but in recent *decades* have both increased and decreased, depending on the specific region and individual compound (O'Rourke et al. 2021).

line 55: I am not sure if 'verify' is the correct word here; maybe 'monitor progress in implementation' woudl be more representative

Good point thank you. Changed wording.

Line 55 - For *instance*, air pollutant and GHG emission inventories can be used within a suite of tools to identify activities that lead to unhealthy air pollution levels or to inform *the design of* policies intended to mitigate air pollution or climate change.

line 63: here and above few references woudl be useful. for example in line 63 a refernce to EPA (when referring to the US) and the EEA/EMEP guidebook and LRTAP Convention (when refering to Europe) could be mentioned ; see also general comments above

We have added these references as suggested. Please also see our comment above on the discussion we've added on the LRTAP Convention and the EMEP/EEA Guidebook.

Line 61 - While there is currently no globally consistent framework for reporting national inventories of air pollutant emissions, national and regional inventory activities have been underway since the 1970s in North America *(e.g., US Environmental Protection Agency) and Europe (e.g., EMEP/European Environment Agency)*, with air pollutant inventory activities now well established in many countries around the world.

line 66: 'this pattern'? what pattern? Are the authors referring to strong increase in pollution and then decline driven by introduction of controls? Agree, but the previous sentences mostly speak about role of inventories and various multiraterla efforst...nothing like this happened in CHina and there is no harmonized and officail emission inventory for China even now.

Language has been clarified as follows:

Line 69 - Air pollution control actions *in China are now* also resulting in substantial emissions reductions for some air pollutants (Zheng et al., 2018b), *although a comprehensive official air pollutant emission inventory has not yet been developed.*

line 110-113: It has been the case in the past, but for several countries already now the role of non-combusion emissions (NH3) has been increasing strongly and even in Europe key primary source of PM is biomass for heating rather than fossil fuels. And so I disagree with this statement as a univeral truth.

We agree with the reviewer and thank them for pointing out the issue with our original language. We have updated this sentence as follows.

Line 118 - The main sources of air pollution in each region differ as well, with a much larger fraction of pollutant emissions in lower income countries originating from solid biofuel use and open burning, while *dominant* sources *tend to initially* shift to shift to fossil fuel combustion in high Socio-Demographic Index (SDI) countries (GBD 2019 Risk Factor Collaborators, 2020, McDuffie et al., 2021). *As also shown in Figure 2, there is further shift in higher income countries away from fossil emissions as air pollution policies reduce combustion emissions, but sources such as agricultural activities or wildfires remain constant or are increasing.*

line 120: How relevant is this statement in perspecctive of this paper? How does this relate to harmonization of emisison inventory methods and challenges?

We have addressed this question in the following paragraphs as noted above.

line 131: Suggest adding 'often' beofre 'major' as it is not always the case, especially if we consider that the text is referring to various temporal and spatial scales where locally (or even regionally - say heavy industry, or poorly controlled industrial nd power coal boilers) carbonaceous aerosols might not be the major part of PM2.5.

Changed. We have also edited this sentence in response to a clarification comment from Reviewer 2.

Line 157 - Note that primary aerosol emissions in *national* air pollutant inventories are often enumerated as $PM_{2.5}$ and $PM_{10}$ (*particles smaller than 2.5 and 10 micrometers in diameter, respectively) to relate to health endpoints*, but for inventories intended for use in climate models are *often* enumerated as black carbon (BC) and organic carbon (OC), which are *often* major components of $PM_{2.5}$ *that help to relate particulate emissions to radiative climate impacts (e.g., Klimont et al., 2017).*

line 137-141: This is the section where i think reference to the internationally accepted and used (while not global) EEA/EMEP guidebook and LRTAP process shall be added.

We agree with the reviewer, please see our response above.

line 146-147: Not only in these countries; Europe has undertaken action way beore any systematic officailyl agreed and harmonized inventory systems were introduced. The observation of healtha and ecosystem damage and link with the air pollution was enoiugh to triger action. Of course progress and triggeering more efficient and coordinated action required agreed methods to report emissions.

We thank the author for highlighting this important point, which is true for both Europe and the U.S. as well. We have added the following sentence to this section:

Line 175 - *In fact, even in Europe and the U.S., early air quality policies preceded coordinated inventory efforts, with official national inventory systems developed later to support and strengthen policy activities.* China provides a *more recent and* interesting example in this respect.

line 163: Maybe i missunderstnad the statments here but i believe thqt in CHina or Vietnam (just to give two examples) the observed PM2.5 is also playing that role. But maybe i am missing something here.

Thank you for the comment. We have removed the words "In these countries", since, as the referee correctly points out, observed concentrations are used to measure compliance in a

wide range of countries. (The remainder of the paragraph refers to the detailed production of emission inventories and their regular use in models and regulatory processes, which is generally confined to higher income countries.)

Line 194 - *The* ultimate regulatory end points are measured pollutant concentrations, which are required to be below specified thresholds for a designated period for a region to be considered in regulatory compliance.

Further, this whole paragraph coudl benefit from refernces to some of the regulations, e.g, NEC Directive in Europe, and also for other regions there are good referneces to established regulatory and enforcement frameworks. For some of the less informed reads it might be useful inforamtion.

We have added references to this section to the history of air pollution legislation in the US, Europe, and Japan.

Line 211 - In Europe, country-level emissions targets have been negotiated under the Gothenburg protocol, making inventory estimates central to this process as well *(Kuklinska et al., 2015; Sliggers and Kakebeeke, 2004; Hashimoto, 1989).*

from line 173: as above, few more refernces woudl be useful, especially to the the Air Convention, since Gothenburg Protocol is referred to in the text.

We have added references as noted above, including a reference to the "25 years of the Convention on Long-range Transboundary Air Pollution"

line 281: Here at the end it woudl be useful to add refernce to Kigali Amendment to the Monteral Protocol since there are specific regioanl/coutnry requirements given there.

We agree with the reviewer that the Kigali Amendment to the Montreal Protocol includes country-specific requirements for those who have ratified. However, the Kigali amendment is focused on phasing down the consumption and production of F-gases and does not require reporting of F-gas emissions, other than HFC-23, which is already a requirement under UNFCCC reporting (recently expanded to developing countries). As the Kigali Amendment is not focused on emissions reporting, we have not included a reference to it in this manuscript.

In response to Dr. Western's comments, we edited this section to clarify that we were attempting to make the point that in comparison to other GHGs, the total anthropogenic emissions of F-gases are relatively easier to derive from direct atmospheric concentration measurements, since F-gases have no natural sources.

Please see our more detailed response above to Dr. Western.

line 283: This paragrpah begins with a statement that I do not agree with. Interntional reporting requirements within the EU and also within the UNECE LRTAP Convention have triggered and actualyl insitutionalized regular update of methods and national emission reporting.

We understand the point, although note that more generally air pollutant emissions reporting has not been strongly influenced by international reporting requirements. Indeed, some developed countries still do not produce complete, official air pollutant emission inventories (e.g. Japan).

However, we have clarified the wording to emphasize that this influence for GHG inventories is 1) global, and b) has been a driving influence from the very start of reporting. This sentence now reads:

Line 354 - In contrast with air pollutant inventories, the use and development of GHG emission inventories has been strongly influenced *globally* by international reporting requirements *from the start of national GHG inventory reporting in the early 1990s.*

Line 294: As above, this last sentence in this paragrpah is not correct. The air pollution inventories reported within the UNECE COnvention are often updated also for historical years and it is regualrly documented in annual submissions and informative reports submitted with it.

Text changed as follows:

Line 366- We note that this sort of past re-estimation is not *always conducted* for air pollutant emissions, *where it is not uncommon to find discontinuities in time series due to methodology changes over time.*

line 372: 'improve air quality coudl unmask' - in this particular statement it owudl approrpaite to make it clear it refers to PM2.5 as for ozone reduction (as the authros discuss further in the paper) the picture is might be different.

Thank you for this comment, we have adjusted the sentence accordingly to be more specific that it is the net aerosol components that are cooling. (Tropospheric ozone is included as a GHG.)

Line 447 - Therefore, while atmospheric aerosols are estimated to have a net cooling effect on the climate, reducing emissions *to reduce particulate matter concentrations* and improve air quality could "unmask" GHG warming that had previously been offset (Lelieveld et al., 2019).

line 454: I think a refernce to the methods used and established in Europe and beyond could be referred to here as they are harmoinized and broadly available and in fact applicable with some exceptions; the global GHG inventory and default emission factors proposed in the IPCC methods are also strongly limted for some GHGs and particular sectors and regions.

This is a good point. We have added this sentence.

Line 540 - *Recent revisions in EMEP/EEA guidance which better harmonizes air pollutant and GHG reporting could be useful in this context, although this guidance is sometimes limited with respect to technologies and practices not common in Europe.*

line 519: But is the large naumber of metrics the issue? I think the problem is rather the problem that one needs a value judgment with respect to why a particualr metric is used as their choice will strongly affedct the relevance of species, sources and therefore policy message.

We agree with the reviewer and have clarified this section as follows. We have also updated this section at the request of Reviewer 2.

Line 621 - In contrast, for air pollutants, it was recommended at a recent IPCC expert meeting (IPCC, 2018) that air pollutant emissions in an SLCF context *continue to be reported in absolute units of mass for each individual compound and not* be aggregated into $CO_2$-*equivalents, as is recommended for GHGs. This is in part* due to their much shorter atmospheric lifetimes (days to weeks vs *decades to centuries) and complex climate and air quality feedbacks, as well as their much larger uncertainties in radiative forcing compared to well-mixed GHGs. These differences* translate to a *very* large range of potential comparison metrics *that would, in essence, require value judgements as to what impact or effect should be represented.*

line 535: reference to MOVES was made also earlier, one coudl also mention other tools like COPERT for Europe, also mobile sources method used widely in Europe and elsewhere. In fact also MOVES is used in some other countries.

Thank you, this is a good point. We have added:

Line 642 - As discussed in Section 2, air pollutant inventories have not historically incorporated emission uncertainty estimates and one challenge for the forthcoming SLCF Methodology Report will be specifying uncertainty guidance for the wide variety of methods used for air pollutant inventory production, which range from complex models (e.g., *for mobile sources: MOVES in the USA and elsewhere, EMFAC in California, and COPERT in Europe)* to direct emissions monitoring.

line 541: I mentioned this before in the general section; i think this discussion divers a little from the main purpose of the paper. These different needs will always remain and there is no need (and it is not feasible) to develop a 'one for all' inventory.

Please see our previous response in the general comments above. We have edited the text throughout section 5 and 6 to highlight that different levels of harmonization may be appropriate for different countries and that there is no single uniform approach to inventory development for all purposes. However, given limited resources for inventory development at the national level, there may still be benefits from varying degrees of coordinated efforts.

line 845: With changing SO2 emissions role of NH3 in formation of ammonium particulkes will be (is) changing

We have edited to add this point, and also in response to other comments, to read:

Line 1071 - *This metric is approximate given* uncertainties and spatial variation in nearly all of these conversion factors, *which will also change over time. For example, the role of NH3 in particle formation is increasing as SO2 emissions decrease.* The globally representative values used here serve to provide a rough estimate of how much particulate matter is produced by emissions of each species and allow us to provide an estimate of the fraction of particulate matter originating from fossil/industrial sources as shown in Figure 2. *While a more detailed calculation would change some details of the figure, we are highlighting here the large-scale features and general trends over time.*

line 863: In fact the paper that is quoted in this paper (Klimont et al 2017) does include mineral PM2.5 at a global level and even national estimates can be derived from the gridded global files. Additionally, maybe relevant, Philip et al (https://doi.org/10.1088/1748-9326/aa65a4) discusses also the role of fugitive and miner dust.

Good point and thank you for the reference. We have added the following text:

Line 1090 - *Mineral $PM_{2.5}$ emissions are implicitly included in most air pollutant inventories as part of $PM_{2.5}$ in global and country level inventories, but generally are not explicitly estimated as a separate category. Because of this mineral $PM_{2.5}$ emissions are not included in most model studies to date (Philip et al., 2017).*

line 875: Maybe adding a refer to Gomez et al (2022; https://www.nature.com/articles/s41467-021-27624-7 ) might be useful

We thank the referee for the reference, which is useful work. We have actually added this reference to the main text, as an example of a coordinated approach to examining joint reduction of GHG and air pollutant emissions.

Line 542 - *Better coordinated inventory development could also facilitate consideration of strategies to jointly reduce both air pollutant and GHG emissions from specific sectors, such as waste (Gómez-Sanabria et al., 2022).*

**RC2: 'Comment on acp-2021-1059', Anonymous Referee #2, 14 Mar 2022**

In this perspective article, the authors present an overview of how GHG and air pollutant emission inventories have developed in the past motivated by different impacts, uses and mitigation goals, highlight challenges and benefits of coordinated GHG and air pollutant emissions inventory development and provide recommendations for such a coordinated reporting within the context of the UNFCCC and IPCC frameworks. Although aerosols, ozone and their precursors gases were recognized by IPCC as climate forcing agents as early as the first assessment report, no effort has been made thus far to define a globally consistent framework for reporting national inventories of short-lived climate forcers. This is partly also due to the fact that SLCFs (except methane which is also a well-mixed greenhouse gas) are not currently part of the mitigation goals. The historical perspectives and recommendations made in this article will be useful for informing the development of the new SLCF emissions reporting guidance initiated by the IPCC.

We thank the reviewer for their helpful comments. We have responded accordingly below, particularly to clarify the definitions we are using for GHGs, air pollutants, and SLCFs throughout this manuscript. These definitions are intended to align with the existing IPCC GHG methodological guidelines. As methane methodologies are already included in existing IPCC GHG Guidelines, we have focused this paper on air pollutants that are direct and indirect climate forcers. Therefore, we have adopted the terminology of 'air pollutant short-lived climate forcers' or 'air pollutant SLCFs'. This terminology is intended to clarify that this definition does not include methane, nor does this include other types of air pollutants (such as toxic chemicals). We have also incorporated additional references where requested and clarified other areas that caused questions or concern. These changes have greatly improved the quality of the manuscript. Our responses to specific comments are below.

Some specific comments that authors may consider:

L28: Mannuci et al only highlight health effects of PM. Please cite studies that also consider ozone and ecosystem effects of air pollution.

We have added two references on the health and ecosystem impacts of ozone.

Line 25 - For example, emissions of short-lived air pollutant species such as sulfur dioxide ($SO_2$), carbonaceous aerosol, and other products of incomplete combustion *or fugitive sources* cause enhanced levels of fine particulate matter and surface level ozone ($O_3$), both of which are harmful to human health and ecosystems (Mannucci et al., 2015, *Malley et al., 2017, EPA, 2020).*

L29: CH4 is considered short-lived (IPCC AR6 chapter 6) and is much shorter lived than well-mixed greenhouse gases such as CO2 and N2O.

We agree with the reviewer that CH4 is much shorter lived than CO2 and N2O. However, in this section, our intent is to compare the atmospheric lifetime of CO2 and CH4 to the relatively shorter lifetimes of air pollutants (discussed in the earlier section), not to each other. We now also clarify in all the relevant sections, the terminology used in this paper to distinguish GHGs (CO2, CH4, N2O, F-gases), air pollutants, and what we're now calling 'air pollutant short-lived climate forcers'. These distinctions have been chosen to align with current UNFCCC/IPCC emission reporting guidelines and existing air pollutant inventories. We have made the relevant changes throughout the manuscript.

Line 29 -  In contrast, emissions of greenhouse gases (GHG), such as methane (CH4) and carbon dioxide (CO2) are *relatively* longer-lived in the atmosphere and alter the Earth's radiative balance, termed radiative forcing, leading to anthropogenic climate change (Myhre et al., 2013).

Line 88 - As a follow-on to the IPCC GHG methodology guidelines and to address uncertainties in *the emissions and impacts of air pollutant short-lived climate forcers (SLCFs)*, the IPCC recently directed its Task Force on National Greenhouse Gas Inventories (TFI) to produce an IPCC methodology report on *SLCFs* (Decision IPCC-XLIX-7, May 2019).

Line 102 - In the *fourth* section, we discuss unique characteristics of "*air pollutant* SLCF inventories" as well as approaches and considerations for quantifying the net health and climate impacts of these compounds.

Line 435 - In this section, *we instead focus* on those SLCFs t*hat impact both air quality and climate, called 'air pollutant SLCFs'*. While the term "air pollutant" can include additional categories of emissions such as heavy metals and toxic compounds (e.g., benzene and perchloroethylene)  that are "known or suspected to cause cancer or other serious health effects" (called Hazardous Air Pollutants in the U.S.)[7],  *we instead are* discussing *the specific* air pollutant SLCF *compounds that will be* considered *by the upcoming* IPCC *SLCF Methodology Report. These* include BC and OC, as well as SO2, NH3, NOx, CO, and NMVOCs, which each uniquely impact the formation of particulate matter and tropospheric O3.

Line 472 - An additional complication is that the forcing effects of individual *air pollutant* SLCFs are much more uncertain than those of GHGs.

Line 725 - "To reduce the development burden of a new national *air pollutant* SLCF-focused inventory, …"

Line 770 - …and the benefits and challenges of coordinating joint GHG and air pollutant emission estimates through the development of an *air pollutant* SLCF inventory."

Line 1039 - Figure 1 - Figure caption -  Schematic diagram illustrating the main characteristics of greenhouse gas (GHGs) vs air pollutant inventories, also highlighting overlap in characteristics inventories of *Air Pollutant* Short-Lived Climate Forcers (SLCF). As discussed in the main text, an

*"Air Pollutant* SLCF inventory" is largely a matter of context rather than a distinct category of inventory data. Note also that all *the* air pollutants *and some of the GHGs* listed above are considered SLCF compounds.

L36: Naik et al (2021) should be cited as Szopa et al (2021)

We have made no change. Naik et al., 2021 is the recommended reference format in Chapter 6 of the 2021 IPCC Report.

L39: Replace air pollutant emissions with precursor emissions
Changed

Line 39 - In addition, O3 in the troposphere, enhanced in the presence of sunlight and *precursor* emissions, is also damaging to both environmental and human health, and is a GHG that contributes to anthropogenic climate change.

L124-128: should be revised to: Inventories of air pollutants include emissions of compounds that are directly hazardous to human health (e.g., nitrogen oxides (NOx = NO2 + NO)) or produce secondary pollutants with detrimental health and ecosystem effects (e.g., tropospheric ozone). These inventories generally include ozone precursor gases such as carbon monoxide (CO), non-methane volatile organic compounds (NMVOCs), and NOx, and primary aerosols (including BC, OC) as well as aerosol precursor gases such as ammonia (NH3), and SO2

We appreciate the reviewer's suggestion for the re-phrasing of these two sentences. Besides the re-ordering of some information, the main difference appears to be the recommendation that we highlight BC and OC as two components of primary aerosol. In the sentences starting on line 157, however, we note that air pollutant inventories may not include BC and OC specification, but instead only report total PM2.5. Therefore, to maintain consistency with that section we have largely retained the original language here, with re-phrasing to help improve readability.

Line 149 - Inventories of air pollutants include emissions of compounds that are directly hazardous to human health, such as nitrogen oxides ($NO_x$ = $NO_2$ + NO) and primary aerosol. These inventories also generally include *precursors to tropospheric O3 and secondary particulate matter*, *such as* gaseous ammonia ($NH_3$), carbon monoxide (CO), non-methane volatile organic compounds (NMVOCs), and $SO_2$.

L132-135: There seems to be a point of confusion about the level of direct PM2.5 emissions and their effect on climate. Can some references be given here?

In this section, our intent was not to discuss the level of impact primary aerosol emissions have on the climate, but rather highlight that aerosol emissions can be included in different forms in inventories that are intended for air quality purposes compared to those intended for use in

climate and atmospheric chemistry models. To clarify this distinction, we have edited this sentence as follows and added a reference:

Line 157 - Note that primary aerosol emissions in *national* air pollutant inventories are often enumerated as $PM_{2.5}$ and $PM_{10}$ (*particles smaller than 2.5 and 10 micrometers in diameter, respectively) to relate to health endpoints*, but for inventories intended for use in climate models are *often* enumerated as black carbon (BC) and organic carbon (OC), which are *often* major components of $PM_{2.5}$ *that help to relate particulate emissions to radiative climate impacts (e.g., Klimont et al., 2017).*

L135: replace "as as well" with "as well as"
Changed.

151-153: Can reference be provided here?
We have added a reference to Zheng et al., 2018a, which summarizes national air quality policies during this period, as well as a direct reference to the China State Council report (translated version), which describes the 10 measures introduced to improve air quality in China.

Line 181- Throughout this same time period, there was increased focus on improving air quality in China, with strict controls on electric power plants and, more recently, large industrial boilers, and emission standards for new vehicles *(Zheng et al., 2018a, China State Council, 2013).*

174-176: A reference would be useful here.

References added to this paragraph.

L182-184: Reference please?

These statements are based on our professional work experience. National pollutant inventory calculations are often supported or conducted by consultants and the details are not published in academic references. To provide two examples of the large data requirements for these inventories, we have added references to the U.S. EPA National Emissions Inventory Technical Support Document that describes the methodologies and data used to develop the national air pollutant emission estimates for the U.S. (in support of regulations under the U.S. Clean Air Act) and the EMEP/EEA Guidebook that provides guidance to countries across Europe.

Line 236 - The development of emissions data used as part of regulatory processes in high SDI countries is often supported by ministries and consultancies that specialize in the production of air pollutant emission inventories. Further, these inventories are made possible by the availability of voluminous ancillary data such as statistics on energy consumption, road

networks, vehicle counts along highways, and economic and industrial activity *(e.g., US EPA, 2021a; European Environmental Agency, 2019).*

L211-214: Tracking compliance will likely be a major reason for detailed inventories even if low SDI countries adopt high controls. Unless the fleet moves over to clean cars (e.g., EV), I don't think one could ignore the source.

We thank the reviewer for their comment. We agree that detailed inventories are vital for compliance tracking in the transportation sector for both high and low SDI countries. The point we were attempting to make in this section, is that inventory resources should be prioritized based on dominant sources of air pollution (or those rapidly changing), given the reality of there being limited resource availability for inventory development. We have rephrased this section to highlight the importance of inventories for compliance tracking, but also to clarify that sources should be prioritized based on their relative importance, given limited resources.

Line 263 - Further, *given limited resources*, inventory development will need to be prioritized based on dominant local sources and current trends. While similar in concept to the IPCC key category analysis for GHG emissions (see below), prioritization can be more complex for air pollutants *given their impacts on climate and air quality, complex feedbacks, and dependence on local conditions and rapidly changing sectors and technologies*. For example, China and India are rapidly adopting road vehicle emission standards similar to those in North America and Europe, *which could* result in a faster movement to higher vehicle *emission* control levels in these countries. *This* stands in contrast to the process in the United States and Europe that o*ccurred over decades*, *where* increasingly strict standards proceeded together with technological developments to facilitate improved emission control devices. *While transportation continues to be a dominant source of GHG emissions*, if most vehicles become compliant with higher control standards *and air pollutant emissions from road vehicles are sufficiently reduced over the coming years, this sector* could become less relevant for the development of new air pollution mitigation policies *in these countries and air pollutant inventory activities could be reprioritized to focus on other major emission sources. Though transportation inventories will continue to be important to track compliance and usage rates[2], these rapid technological changes suggest that resources for future air pollutant inventory improvements* may best be focused on sectors and subsectors (*e.g.*, off-road mobile sources in the USA) or specific emitted compounds (*e.g.,* NMVOCs in China) that have *recently* become relatively larger *sources of* air pollution.
* * *
*[2]Tracking usage rates for older vehicles will be increasingly important since once the majority of vehicles are equipped with strong emission controls, road vehicle emissions will be increasingly impacted by super-emitters. Carslaw, D. C., Beevers, S. D., Tate, J. E., Westmoreland, E. J., and Williams, M. L.: Recent evidence concerning higher NOx emissions from passenger cars and light duty vehicles, Atmos Environ, 45, 7053-7063, 10.1016/j.atmosenv.2011.09.063, 2011.*

L214: Insert a "." between 'access Therefore'

Changed

Section2.1 : Why only air pollution impacts are discussed here? Why not the climate change impacts of air pollutants and the uncedrtainty associated with lack of consistent information on their emissions trends, which is a signifcant motivating factor for IPCC to develop reporting methodology for SLCFs.

The reviewer raises an important point regarding the discussion surrounding the climate impacts of air pollutant SLCFs. In the current organization, we provide a detailed discussion of the climate impacts (both direct and indirect) of these compounds in Section 4 (Air Pollutant SLCF) and instead focus Section 2 (Air Pollutant Emissions) on their air quality and health impacts. Therefore, we have adjusted the first sentence in this section to acknowledge the climate impacts and to notify the reader that these are discussed elsewhere.

Line 111 - *For air pollutants, in addition to their climate impacts (discussed in Section 4), l*ong-term exposure to both indoor and outdoor air *pollutants (i.e., $O_3$ and particulate matter)* was attributable to nearly 7 million deaths worldwide in the year 2019, corresponding to the loss of over 213 million disability-adjusted life-years (GBD 2019 Risk Factor Collaborators, 2020).

L249-250: what about methane? Methane's effect on climate is in the short-term

Please see our related responses above. This sentence is another instance where we are comparing the atmospheric lifetime and effects of greenhouse gases relative to those of air pollutants. While methane has short-term impacts relative to CO2, the impacts are still on relatively longer timescales as compared to air pollutant impacts on air quality or climate. We have updated the following sentence to clarify that here we are comparing the time scales of climate change relative to air pollution.

Line 312 - The time horizon for climate change is also *relatively* longer *than for air pollution*, given both the global scope of fossil-fuel consumption and the century to millennial atmospheric time scales of the carbon-cycle.

L251-254: This statement needs to be caveated since methane, a GHG, is short-lived and its concentrations have been helpful in developing top-down emissions inventories

We agree with the reviewer that observationally-based emission estimates of methane can be helpful for both the development of top-down emission inventories and the evaluation of existing bottom-up inventories. In our original text, we included a footnote clarifying this exact point, that GHG concentrations could be used to develop GHG emission estimates. We have made minor modifications to this footnote and its placement in the text to highlight this point more clearly.

Line 316 - Further, in contrast with air pollutants, *while emission estimates derived from* atmospheric concentrations of GHGs *can help inform inventory efforts[3]*, GHG concentrations are not particularly useful as near-term regulatory end-points, as concentrations change slowly and are influenced by short-term variations in the carbon-cycle and atmospheric chemistry.

[3] It is also possible to estimate GHG emissions from *inverse modeling* of concentration measurements, from either s*tationary platforms (e.g., towers, surface stations), mobile platforms (e.g., aircraft), and/or satellite retrievals*. Such inversion estimates can play a role in GHG inventory development, particularly in identifying large sources and identifying regions *or sectors* where inventories may have large biases.

L271: I don't understand this construct – is this ±?
The reviewer is correct, we have provided the +/- uncertainty range presented in the referenced study (Zimmerle et al., 2015). In this case, the uncertainty range is -23% to + 36%. We have edited the text to clarify and also corrected an additional error that +19% should have been reported as -19%.

Line 356 - Zimmerle et al. (2015) estimate, for example, that uncertainty in fugitive $CH_4$ emissions from the transmission and storage of natural gas in the United States i*s between - 23% and +36%* (+30%/*-19%* if unmodeled portions of the sector are included), and that over 36% of emissions from this sector are from a small number of individual sources.

L274: define EDGAR
Changed.

Line 341 - As estimated by the global *Emissions Database for Global Atmospheric Research (EDGAR)* GHG inventory (Minx et al. 2021, Crippa et al. 2021), fugitive $CH_4$ emissions from oil and gas operations were less than 20% of total national $CH_4$ emissions in 86 of the top 100 GHG emitting countries in 2015.

L288: reference?
We added a reference to the IPCC 2006 GHG Guidelines.

Line 361- Emissions reporting is facilitated by detailed templates and methodological guidelines from the IPCC (i.e., IPCC inventory guidelines) *(IPCC, 2006)*.

L376-381: References would be useful here.
We have added two references regarding the chemical feedbacks between O3 and methane and the non-linearities in O3 production from NOx and NMVOCs.

Line 452- Understanding the net impact of reactive gas emissions is even more complex. For example, when considering their net radiative impacts, emissions of $NO_x$, CO, and NMVOCs not only impact atmospheric concentrations of $CH_4$ and $O_3$, but changes in $CH_4$ concentrations will then feedback to impact background levels of $O_3$ (*e.g., Fiore et al., 2008*). In addition, due to non-linearities in atmospheric chemistry, changes in $NO_x$ and NMVOC emissions can either increase or decrease local $O_3$ concentrations, depending on background conditions and the relative emissions of each *(McDuffie et al., 2016)*.

L384-387: Also Szopa et al (2021)
We have added a reference to the 2021 IPCC report to this sentence as the reviewer suggested.

Line 460 - In terms of the global climate, emissions of $CH_4$, CO, and NMVOCs have increased the historical forcing from both $CH_4$ and background tropospheric $O_3$ (Myhre et al., 2013, *Naik et al., 2021*).

L409: reference?
We have added a reference to Box 6.1 in Chapter 6 of the 2021 IPCC report, which describes the need to reduce the complexity of chemical mechanisms to meet the required computational efficiency needs of chemistry climate models. We have also added a reference to Appel et al., 2021, which describes the detailed chemical mechanism in the latest version of the CMAQ model, used for regulatory air quality and health impact analyses in the U.S.

Line 485 - Similar models are also used to examine the environmental and health impacts of these emissions, although models used for these purposes may have more detailed representations of atmospheric chemistry than those used for long-term simulations of climate change *(e.g., Box 6.1 in Naik et al., 2021, Appel et al., 2021)*.

L504-506: reference?
We have added two references to the Global Emissions InitiAtive (GEIA) NMVOC emissions working group page and to Huang, G, et al., 2017 (NMVOC speciation work from the EDGAR group) to support the statement that chemical speciation is currently an active area of research.

Line 610 - Chemical speciation is currently an active area of research *(GEIA: Global Emissions InitiAtive, Huang et al., 2017)* and air pollutant inventory development will need to consider flexible guidance on appropriate emission speciation profiles as well.

L508: equivalant should be equivalent throughout the manuscript
Thank you for catching this typo. Changed throughout manuscript.

L515-519: Allen et al (2022) https://www.nature.com/articles/s41612-021-00226-2 could also be useful here.

We appreciate the reviewer's recommendation, however, in this section we are simply highlighting the recommendation from a previous IPCC meeting that emissions of air pollutant SLCFs should continue to be reported in absolute mass units of each individual compound rather than combined using a CO2-equivalent metric, as is currently done for GHGs. There is a broad range of literature on the subject of common reporting metrics (including Allen et al., 2022), for both GHGs, SLCFs, and the combination of the two, which each disproportionately weight a different policy priority (e.g., climate impacts on varying time scales, local/regional health impacts, etc). A detailed discussion of reporting metrics is outside the scope of this manuscript and we do not want to appear to be endorsing a specific metric by referencing a select subset of these papers. Rather, in this manuscript, we simply recommend that due to the varying atmospheric lifetimes, complex climate and air quality feedback loops, and the regulatory purposes and priorities of air pollutant SLCF inventories, that these emissions continue to be reported on a mass basis for each individual compound. Therefore, we have made minor modifications to this section to clarify our recommendation but have not added the recommended reference so as not to emphasize a particular type of metric. We have also edited this section in response to Reviewer #1.

Line 621 - In contrast, for air pollutants, it was recommended at a recent IPCC expert meeting (IPCC, 2018) that air pollutant emissions in an SLCF context *continue to be reported in absolute units of mass for each individual compound and not* be aggregated into $CO_2$-*equivalents, as is recommended for GHGs. This is in part* due to their much shorter atmospheric lifetimes (days to weeks vs *decades to centuries) and complex climate and air quality feedbacks, as well as their much larger uncertainties in radiative forcing compared to well-mixed GHGs. These differences* translate to a *very* large range of potential comparison metrics *that would, in essence, require judgements as to what impact or effect should be represented.*